# Physiology as Language: Translating Respiration to EEG during Sleep

**Kaiwen Zha** [* 1]   **Chao Li** [* 1]   **Hao He** [* 1]   **Peng Cao** [1]   **Tianhong Li** [1]   **Ali Mirzazadeh** [1]   **Ellen Zhang** [1]
**Jong Woo Lee** [2]   **Yoon Kim** [1]   **Dina Katabi** [1]

## Abstract

This paper introduces a novel cross-physiology translation task: synthesizing sleep electroencephalography (EEG) from respiration signals. To address the significant complexity gap between the two modalities, we propose a waveform-conditional generative framework that preserves fine-grained respiratory dynamics while constraining the EEG target space through discrete tokenization. Trained on 28,394 individuals, our model achieves a 7% Mean Absolute Error in EEG spectrogram reconstruction. Beyond reconstruction, the synthesized EEG supports downstream tasks with performance comparable to ground truth EEG on age estimation (MAE 5.0 vs. 5.1 years), sex detection (AUROC 0.81 vs. 0.82), and sleep staging (Accuracy 0.84 vs. 0.88), significantly outperforming baselines trained directly on breathing. Finally, we demonstrate that the framework generalizes to contactless sensing by synthesizing EEG from wireless radio-frequency reflections, highlighting the feasibility of remote, non-contact neurological assessment during sleep.

## 1. Introduction

Physiological monitoring is central to modern healthcare, particularly for managing neurological and sleep disorders. The electroencephalogram (EEG), in particular, is the gold standard for measuring brain activity. It supports the diagnosis of conditions ranging from epilepsy and narcolepsy to PTSD and depression (Acharya et al., 2015; Newson & Thiagarajan, 2019), while also serving as a biomarker for drug toxicity and brain aging (Van Cott & Brenner, 2003; Iosifescu, 2011; Sun et al., 2019).

*Equal contribution [1]MIT Computer Science & Artificial Intelligence Laboratory [2]Brigham and Women's Hospital, Harvard Medical School. Correspondence to: Kaiwen Zha <kzha@mit.edu>, Chao Li <chaoli@mit.edu>, Hao He <haohe@mit.edu>, Peng Cao <pengcao@mit.edu>.

*Proceedings of the 43$^{rd}$ International Conference on Machine Learning*, Seoul, South Korea. PMLR 306, 2026. Copyright 2026 by the author(s).

However, the clinical utility of EEG is limited by the difficulty of acquisition. Standard polysomnography (PSG) relies on wet electrodes or tight headbands that are labor-intensive to apply and prone to artifacts. Worse, this cumbersome instrumentation can disrupt the very sleep it aims to measure (Agnew Jr et al., 1966). Respiration, by contrast, is far easier to capture, requiring only non-invasive wearables such as breathing belts or even contactless radio-frequency (RF) sensors (Yue et al., 2018). This gap—between the diagnostic value of EEG and the accessibility of breathing signals—motivates a fundamental question: *Can we synthesize high-fidelity EEG from breathing alone during sleep?*

Physiology traditionally treats the pulmonary and neurological systems as distinct domains. Yet growing evidence of respiratory–neurological coupling, particularly during sleep, suggests a deep, often hidden interdependence between them (Zelano et al., 2016; Heck et al., 2017; Kluger & Gross, 2021). This coupling operates through at least three mechanisms. First, breathing acts as a timing signal that modulates cortical excitability (Heck et al., 2017) and aligns with the slow oscillations, spindles, and coupled complexes that form the core building blocks of sleep EEG (Schreiner et al., 2023). Second, respiratory and arousal systems share brainstem regulation: the pre-Bötzinger complex generates respiratory rhythms and projects to the locus coeruleus and mediodorsal thalamus, linking breathing to central arousal control (Yackle et al., 2017). Third, brain states continuously reshape breathing patterns—deep NREM produces slow, regular respiration; REM induces irregularity; arousals trigger transient disruptions—embedding sleep-state information directly into the respiratory waveform.

While this coupling is well-established, the literature does not show that full sleep EEG spectrograms can be reconstructed from respiration alone. However, machine learning routinely uncovers latent physiological relationships that elude traditional analysis: diagnosing Parkinson's from nocturnal breathing (Yang et al., 2022), predicting 130 conditions from a single night of PSG (Thapa et al.), identifying kidney disease and diabetes from retinal images (Zhang et al., 2021), and detecting hyperthyroidism from ECGs (Lin et al., 2024)—all despite the lack of established predictive links. Building on these precedents, we hypothesize that respiratory dynamics encode shared latent information

about brain activity that modern machine learning can extract. If breathing carries a "fingerprint" of brain activity during sleep, then learning a mapping from respiration to EEG could democratize neurological monitoring, enabling scalable, comfortable, and potentially contactless assessment of brain activity.

We adopt the view that physiological time series, like language, can be described by a vocabulary of recurring patterns governed by biological "grammar," and that different signals (e.g., EEG and breathing) may encode overlapping health information using distinct vocabularies. Based on this premise, we introduce a waveform-conditional generative model that translates breathing signals into sleep EEG. The model conditions on continuous respiratory waveforms, while treating EEG as a discrete language: it first learns an EEG vocabulary via tokenization, then trains a Transformer to translate respiration into EEG tokens using a masked-prediction objective. This asymmetric design preserves fine-grained continuous respiratory context while constraining the EEG search space, enabling high-fidelity generation.

We validate our approach at unprecedented scale using 14 sleep datasets spanning 28,394 individuals and 33,919 nights. Breathing-to-EEG translation achieves 7% mean absolute error (MAE), preserving the key spectral and temporal structure of the ground-truth EEG. To assess utility beyond reconstruction, we evaluate the synthesized EEG on three downstream tasks—age regression, sex classification, and sleep staging—and benchmark against models trained on ground-truth EEG and on respiration alone. Across datasets, synthesized EEG approaches ground-truth performance (age MAE 5.0 vs. 5.1 years; sex AUROC 0.81 vs. 0.82; sleep-staging accuracy 0.84 vs. 0.88), while models trained directly on respiration perform substantially worse. This gap suggests that our model can disentangle and amplify brain-specific information latent in respiratory dynamics in a way that raw breathing signals do not.

Finally, we push toward non-invasive EEG monitoring using the MGH dataset, which includes synchronized RF-based breathing, belt-based breathing, and EEG. This unique setting allows us to demonstrate the feasibility of generating meaningful sleep EEG from wireless reflections alone: RF signals reflected off a sleeping body can be translated into EEG spectrograms at an accuracy comparable to those from a breathing belt (MAE of 8% vs. 7%), enabling neurological assessment without physical contact or wearables.

Our contributions are summarized as follows:

- **A new translation task and a waveform-conditional framework:** We introduce cross-physiology translation from the pulmonary to the neurological domain during sleep and show that a waveform-conditional model with a tokenized EEG target enables accurate reconstruction and supports downstream tasks.

- **Feasibility of contactless EEG:** We provide the first demonstration of generating sleep EEG spectrogram from contactless wireless reflections, establishing a pathway toward remote, contact-free neurological assessment.

- **Large-scale clinical validation:** Across a diverse cohort spanning 14 datasets, we show that synthesized EEG is not only visually faithful but also computationally functional, achieving competitive performance on sleep staging, age estimation, and sex classification.

## 2. Related Works

**Masked Generative Modeling.** Our architecture builds on masked generative modeling, where representing data as discrete token sequences enables powerful, transferable models across domains. In NLP, masked reconstruction objectives (e.g., BERT (Devlin et al., 2019), MASS (Song et al., 2019), BART (Lewis et al., 2020)) support both representation learning and sequence-to-sequence generation. This paradigm has also been extended to vision, where tokenized images with masked prediction enable scalable generation, as shown by MaskGIT (Chang et al., 2022), Fluid (Fan et al., 2024) and DREAM (Li et al., 2026). We extend these ideas to waveform-conditional generation across physiological domains, translating continuous respiratory waveforms into tokenized EEG for cross-domain physiological synthesis.

**Cross-Modal Physiological Learning.** Prior multimodal learning on physiological data largely focuses on fusing signals for discriminative tasks (Thapa et al.; Zhang et al.; Abbaspourazad et al.; Deldari et al.; Fang et al.). More recently, researchers have explored generating one physiological modality from another to impute missing data or improve monitoring. However, existing work translates between modalities within the same physiological domain (e.g., neurological or cardiovascular), where signals are tightly coupled and share substantial information. In the neurological domain, NT-ViT (Lanzino et al., 2024) and diffusion-based methods (Ma et al., 2026) translate EEG to fMRI to estimate spatially resolved hemodynamic activity from electrical recordings. In the cardiovascular domain, deep generative models translate PPG to ECG (Sarkar & Etemad, 2021; Lan, 2023; Tang et al., 2023; Ezzat et al., 2024) and PPG to continuous arterial blood pressure (ABP) waveforms (Ibtehaz et al., 2022). In contrast, we are the first to demonstrate translation across physiological domains.

**Extracting Physiological Signals from Wireless Reflections.** There is growing interest in passive health monitoring, where a low-power radio device (akin to a WiFi router) emits signals and analyzes their reflections to infer physiological measurements without wearables (Islam, 2022). Prior work has used this paradigm to estimate a range of physiologi-

cal and behavioral signals, including respiration (Yue et al., 2018; Zhang et al., 2023), heart rate (Adib et al., 2015), pose (Zhao et al., 2018), stress(Ha et al., 2021), activity (Liu et al., 2020), and even sleep stages (He et al., 2025). We extend this work by demonstrating, for the first time, that sleep EEG can be inferred from wireless reflections alone, enabling contact-free neurophysiological monitoring.

## 3. Method

We propose a cross-physiology translation framework to synthesize high-fidelity sleep EEG directly from nocturnal breathing signals. This task presents a fundamental challenge: the significant *complexity gap* between the input and the target. Respiratory signals are relatively simple, low-frequency waveforms driven by mechanical pulmonary effort, whereas EEG signals are complex, high-frequency, and stochastic representations of neurological activity.

We address this challenge by employing *asymmetric* processing strategy for the two modalities to align their information density. For the source breathing signal, we prioritize the preservation of fine-grained context by encoding the raw waveform directly into a sequence of dense, continuous embeddings. In contrast, for the target EEG, we constrain the high-dimensional search space by converting the signal into a time-frequency spectrogram and discretizing it via a learned codebook of EEG patterns. This asymmetric embedding strategy enables us to leverage a transformer-based masked generative model to effectively translate the detailed, continuous respiratory context into the semantic, discrete vocabulary of brain activity. Our model pipeline is illustrated in Fig. 1.

### 3.1. Asymmetric Embedding

**Source: Raw Waveform Embedding.** We employ a minimal projection layer to preserve detailed physiological information in the raw waveform. Specifically, we divide the raw breathing signal $X_{\text{RESP}}$ into a sequence of non-overlapping 4-min segments and linearly map each segment into a continuous embedding space. This results in a sequence of continuous *breathing tokens* that allows the subsequent Transformer to attend to fine-grained respiratory dynamics. This linear layer is trained together with the subsequent transformer.

This raw representation is critical. Our experiments revealed that utilizing heavy breathing encoders or time-frequency conversions significantly degraded performance, likely by filtering out subtle morphological cues or over-compressing the temporal context required to infer brain states.

**Target: Discrete Spectrogram Tokenization** While the input must be preserved in raw form, the target EEG signal $X_{\text{EEG}}$ is noisy and its structure is distributed across frequencies, making direct waveform generation challenging. To

make the search space tractable and capture the underlying EEG structure, we follow a two-step process:

*1. Spectral Transformation*: We first convert the time-series single-channel EEG signal into a time-frequency spectrogram using the multitaper method (Prerau et al., 2017). This approach provides Power Spectral Density (PSD) estimates with high spectral resolution and minimal leakage, capturing essential sleep EEG features such as posterior dominant rhythms (8–12 Hz), sleep spindles (12–16 Hz), and slow-wave activity (0.5–1.5 Hz) (Prerau et al., 2017), as shown in Fig. 3. We process the data using a window size of 30 seconds; this duration is chosen to align with the standard epoch length defined by the AASM Manual for the Scoring of Sleep and Associated Events (Berry et al., 2012).

*2. Vector Quantization (VQ)*: We further compress the spectrogram into a sequence of discrete *EEG tokens* using a VQGAN (Esser et al., 2021). This process effectively shrink the target search space, allowing the translation task to be a tractable classification problem over a finite vocabulary of physiological patterns instead of a complex regression.

As shown in Fig. 2, the VQGAN is composed of a encoder $E$, a decoder $D$ and a learnable codebook $\mathcal{Z} = \{z_k\}_{k=1}^{K} \subset \mathbb{R}^d$. The encoder takes a spectrogram and produces a latent feature grid $\hat{z} = E(X_{\text{EEG}}) \in \mathbb{R}^{h \times w \times d}$, and then each feature vector is replaced by its nearest neighbor from the codebook, $z_q = \arg\min_{z_k \in \mathcal{Z}} ||\hat{z} - z_k||_2$, to obtain the quantized token sequence $z_q$. The decoder converts these tokens back to a spectrogram such that $\hat{X}_{\text{EEG}} = D(z_q) \approx X_{\text{EEG}}$.

We design the encoder's downsampling factor such that each token corresponds to a patch of approximately 4 Hz in frequency and 4 minutes in time. This resolution is strategically chosen to align the latent representation with the physiological semantics of sleep. The 4 Hz frequency resolution naturally segments the spectral energy into canonical EEG bands, isolating Delta (0-4 Hz), Theta (4-8 Hz), Alpha (8-12 Hz), and Sigma (12-16 Hz) into distinct latent rows. The 4-minute temporal resolution aggregates sufficient context to capture stable meso-scale sleep states.

The VQGAN is pretrained separately. In addition to the standard VQGAN objective (Esser et al., 2021), which jointly optimizes reconstruction fidelity, codebook commitment, and adversarial losses, we incorporate a correlation regularization term that explicitly maximizes the correlation between reconstructed spectrograms and the ground-truth targets. This encourages the tokenizer to preserve intrinsic physiological variability and spectral morphology, rather than converging to overly smooth or "average" patterns.

### 3.2. Waveform-Conditional Generative Translation

With the asymmetric representations established—long-context raw embeddings for breathing and discrete semantic

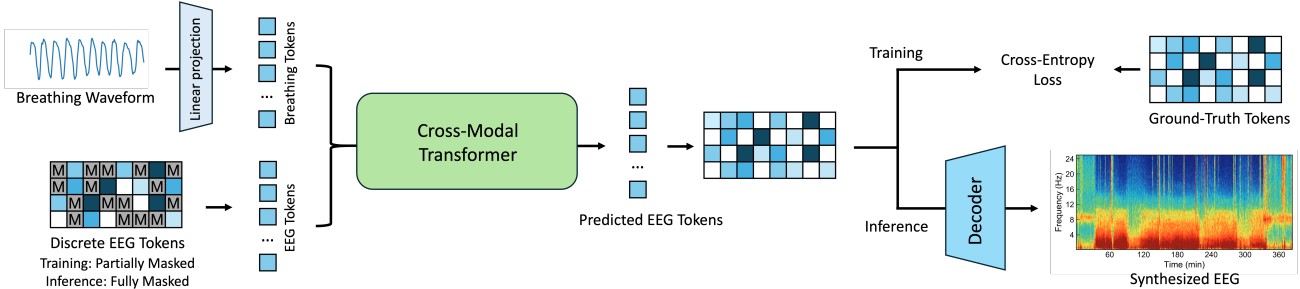

*Figure 1.* **Model Pipeline.** The model synthesizes sleep EEG from nocturnal breathing signals using an asymmetric embedding strategy to bridge the gap between modalities. The source breathing signal is processed as a raw waveform with a linear projection, while the target EEG is converted into discrete tokens by spectral transformation and vector quantization. A transformer-based model learns to translate the continuous respiratory context into the discrete neurological states using a masked generative modeling objective. During inference, the model predicts the full sequence of EEG tokens from breathing alone, which are then decoded into an EEG spectrogram.

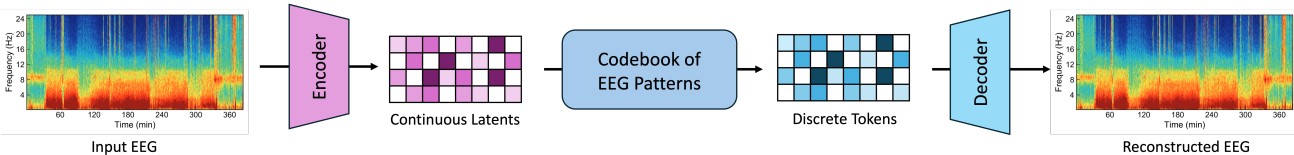

*Figure 2.* **Vector Quantization for EEG.** The EEG spectrogram is discretized into tokens via a codebook of distinct EEG patterns. The resolution of the token (4 Hz × 4 minutes per token) is chosen to align with the physiological semantics of sleep EEG.

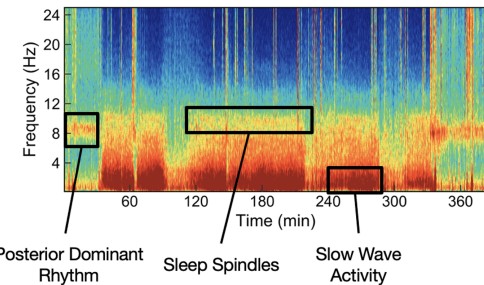

*Figure 3.* **An example of sleep EEG spectrogram.** Key sleep patterns are annotated, including the posterior dominant rhythm (low-frequency activity around 8-12 Hz), sleep spindles (short bursts of 12-16 Hz activity), and slow-wave activity (prominent low-frequency oscillations below 4 Hz).

tokens for EEG—we formulate the synthesis as a sequence-to-sequence translation task. We employ a Transformer-based architecture and leverage a masked generative modeling approach (Chang et al., 2022; Li et al., 2023).

During training, the transformer processes a concatenated sequence of breathing tokens and partially masked EEG tokens. We add learnable positional embeddings to preserve structure: 1D temporal embeddings for the breathing sequence and 2D embeddings (encoding time and frequency) for the EEG tokens. The model operates over this joint sequence to reconstruct the discrete tokens at masked positions, optimized via a cross-entropy loss. The masking ratio $\gamma$ is sampled from a truncated Gaussian distribution ($\mu = 0.55$, clipped to $[0.5, 1.0]$) following Li et al. (2023). This variable masking schedule exposes the model to di-

verse difficulty levels, fostering robust contextual learning between the pulmonary and neurological modalities.

During inference, we synthesize EEG from a new breathing input by conditioning the model on breathing tokens concatenated with a *fully masked* EEG sequence ($\gamma = 1.0$). The predicted tokens are subsequently decoded by the frozen VQGAN to yield the final EEG spectrogram. Our experiments demonstrate that this masked generation strategy achieves superior reconstruction fidelity compared to sequential autoregressive generation.

## 4. Experiments

### 4.1. Evaluation Details and Downstream Tasks

**Reconstruction.** We evaluate EEG reconstruction by comparing the synthesized EEG spectrograms to their ground truth counterpart. We use two metrics: (1) **Mean absolute error (MAE)** computed on spectrogram values normalized to $[0, 1]$ and (2) **Signal-to-noise ratio (SNR)**, measuring signal power relative to reconstruction error: $\text{SNR} = 10 \log \frac{\|S\|^2}{\|S-\hat{S}\|^2}$. We report both metrics for the full EEG spectrogram and for each canonical frequency bands: $\delta$ (0–4 Hz), $\theta$ (4–8 Hz), $\alpha$ (8–12 Hz), $\sigma$ (12–16 Hz), and $\beta$ (12–32 Hz). Since no prior models reconstruct EEG from breathing, this task has no existing baselines for comparison. Thus, our evaluation focuses on reconstruction accuracy estimated via the above metrics.

*Table 1.* **Mean Absolute Error (MAE) of EEG reconstruction from breathing signals across datasets and frequency bands.** (a) Internal datasets are used during model development, while (b) External datasets are held out for evaluation only. Values are reported as mean ± standard deviation of MAE. **BB** denotes breathing from a wearable belts and **RF** denotes breathing from wireless reflections. Datasets marked with a dagger ($^\dagger$) include data from multiple sleep lab visits.

*(a) Internal datasets*

| Dataset | Source | Overall | $\delta$ (0–4Hz) | $\theta$ (4–8Hz) | $\alpha$ (8–12Hz) | $\sigma$ (12–16Hz) | $\beta$ (12–32Hz) |
|---|---|---|---|---|---|---|---|
| BWH | BB | 0.056 ± 0.017 | 0.062 ± 0.023 | 0.053 ± 0.022 | 0.054 ± 0.020 | 0.052 ± 0.018 | 0.056 ± 0.018 |
| SHHS$^\dagger$ | BB | 0.057 ± 0.020 | 0.057 ± 0.022 | 0.051 ± 0.022 | 0.058 ± 0.021 | 0.055 ± 0.024 | 0.058 ± 0.023 |
| MROS$^\dagger$ | BB | 0.066 ± 0.020 | 0.066 ± 0.025 | 0.062 ± 0.024 | 0.062 ± 0.022 | 0.060 ± 0.021 | 0.068 ± 0.020 |
| CHAT$^\dagger$ | BB | 0.067 ± 0.048 | 0.065 ± 0.045 | 0.069 ± 0.050 | 0.064 ± 0.052 | 0.066 ± 0.052 | 0.067 ± 0.050 |
| CCSHS | BB | 0.066 ± 0.015 | 0.072 ± 0.020 | 0.064 ± 0.021 | 0.064 ± 0.018 | 0.066 ± 0.016 | 0.067 ± 0.017 |
| NCHSDB | BB | 0.081 ± 0.036 | 0.080 ± 0.041 | 0.081 ± 0.040 | 0.076 ± 0.039 | 0.079 ± 0.038 | 0.083 ± 0.036 |
| P18C | BB | 0.104 ± 0.071 | 0.107 ± 0.072 | 0.102 ± 0.076 | 0.107 ± 0.080 | 0.102 ± 0.076 | 0.103 ± 0.070 |
| STAGES | BB | 0.089 ± 0.050 | 0.096 ± 0.054 | 0.086 ± 0.055 | 0.088 ± 0.055 | 0.087 ± 0.054 | 0.089 ± 0.051 |
| MGH | BB | 0.069 ± 0.026 | 0.074 ± 0.030 | 0.068 ± 0.032 | 0.070 ± 0.031 | 0.064 ± 0.029 | 0.067 ± 0.028 |
| MGH | RF | 0.076 ± 0.027 | 0.085 ± 0.038 | 0.075 ± 0.035 | 0.075 ± 0.031 | 0.070 ± 0.029 | 0.074 ± 0.027 |
| **Average** | | 0.068 ± 0.036 | 0.070 ± 0.039 | 0.065 ± 0.040 | 0.067 ± 0.039 | 0.065 ± 0.039 | 0.069 ± 0.037 |

*(b) External datasets*

| Dataset | Source | Overall | $\delta$ (0–4Hz) | $\theta$ (4–8Hz) | $\alpha$ (8–12Hz) | $\sigma$ (12–16Hz) | $\beta$ (12–32Hz) |
|---|---|---|---|---|---|---|---|
| WSC | BB | 0.059 ± 0.016 | 0.068 ± 0.022 | 0.053 ± 0.020 | 0.058 ± 0.018 | 0.055 ± 0.017 | 0.059 ± 0.017 |
| CFS | BB | 0.073 ± 0.019 | 0.089 ± 0.031 | 0.080 ± 0.030 | 0.070 ± 0.022 | 0.067 ± 0.019 | 0.070 ± 0.020 |
| MESA | BB | 0.073 ± 0.044 | 0.079 ± 0.053 | 0.072 ± 0.053 | 0.071 ± 0.050 | 0.069 ± 0.046 | 0.073 ± 0.043 |
| SOF | BB | 0.067 ± 0.020 | 0.075 ± 0.028 | 0.064 ± 0.026 | 0.063 ± 0.024 | 0.060 ± 0.022 | 0.067 ± 0.020 |
| UMASS | RF | 0.073 ± 0.024 | 0.092 ± 0.029 | 0.078 ± 0.029 | 0.080 ± 0.029 | 0.082 ± 0.032 | 0.066 ± 0.024 |
| **Average** | | 0.067 ± 0.030 | 0.075 ± 0.038 | 0.064 ± 0.038 | 0.064 ± 0.034 | 0.062 ± 0.032 | 0.066 ± 0.030 |

**Downstream Tasks.** We further evaluate the utility of the synthesized EEG spectrograms on three downstream tasks: age estimation, sex detection, and sleep-stage prediction. We compare models trained on synthesized EEG against two baselines: (1) **GT-EEG**: models trained on ground-truth EEG and (2) **Breathing**: models trained directly on raw breathing signals. This comparison quantifies the benefit of synthesizing EEG over using breathing alone, and how closely performance approaches that of ground-truth EEG.

All models take a full night of data as input. Age prediction is formulated as regression and trained with mean squared error (MSE) loss. Sex prediction is formulated as a binary classification and trained with binary cross-entropy loss. Sleep-stage prediction is posed as sequence labeling, predicting one label (Wake, Light, Deep, or REM) per 30-second segment and optimized with cross-entropy loss.

Models operating on EEG use a ViT/Ti backbone ($\sim$5.7M parameters). For breathing baselines, we compare (1) a ResNet-style model He et al. (2025) (current SOTA) and (2) a hybrid transformer that prepends a convolutional encoder to the ViT/Ti backbone. The latter controls for architectural capacity, isolating gains from the synthesized EEG.

For the prediction heads, we adapt the pooling strategy to the task level. For recording-level tasks (age and sex prediction), we apply global average pooling over the ViT token embeddings to obtain a single representation. Conversely, for sleep-stage prediction, we process the full sequence of embeddings to generate per-segment labels. Performance is reported using Mean Absolute Error (MAE) for age, AUROC for sex, and accuracy for sleep staging.

**Training and Cross Validation.** We perform 4-fold, patient-wise cross-validation on the internal training datasets, assigning all nights from a given participant to the same fold. In each run, the model is trained on three folds and evaluated on the held-out fold, yielding subject-disjoint splits and preventing leakage from repeat visits. We use the same fold partitions for both VQGAN pretraining and translation model training.

**Datasets.** We analyze a large-scale sleep corpus comprising 33,919 nights from 28,394 individuals (mean age 52.6 years; SD 25.1; range 3–102 years). Participants are 44.7% female, and 72.8% identify as White. The corpus aggregates 14 distinct sleep datasets: 9 are used for training and cross-validation, and 5 are held out as external test sets to evaluate out-of-distribution generalization. All 14 datasets include

*Table 2.* **Signal-to-noise ratio (SNR) of EEG reconstruction from breathing signals across datasets and frequency bands.** (a) Internal datasets used during model development and (b) external datasets held out during training for evaluation. Values are reported as mean ± standard deviation of SNR. **BB** denotes breathing belts and **RF** denotes wireless breathing sensors. Datasets marked with a dagger ($^\dagger$) include data from multiple sleep lab visits.

*(a) Internal datasets*

| Dataset | Source | Overall | $\delta$ (0–4Hz) | $\theta$ (4–8Hz) | $\alpha$ (8–12Hz) | $\sigma$ (12–16Hz) | $\beta$ (12–32Hz) |
|---|---|---|---|---|---|---|---|
| BWH | BB | 14.9 ± 1.9 | 17.0 ± 2.4 | 17.2 ± 2.8 | 16.6 ± 2.6 | 15.9 ± 2.4 | 13.6 ± 2.0 |
| SHHS$^\dagger$ | BB | 15.4 ± 1.8 | 19.0 ± 2.3 | 18.5 ± 2.6 | 17.0 ± 2.5 | 16.3 ± 2.4 | 13.8 ± 1.9 |
| MROS$^\dagger$ | BB | 14.1 ± 1.9 | 17.3 ± 2.4 | 16.3 ± 2.6 | 15.9 ± 2.6 | 15.0 ± 2.3 | 12.6 ± 2.0 |
| CHAT$^\dagger$ | BB | 14.5 ± 3.7 | 19.2 ± 3.7 | 17.0 ± 3.9 | 15.5 ± 4.1 | 14.0 ± 4.3 | 12.1 ± 4.3 |
| CCSHS | BB | 14.2 ± 1.9 | 17.7 ± 2.2 | 16.6 ± 2.7 | 15.3 ± 2.5 | 14.1 ± 2.1 | 12.3 ± 1.9 |
| NCHSDB | BB | 12.5 ± 2.6 | 16.7 ± 3.4 | 14.8 ± 3.2 | 13.4 ± 3.1 | 12.1 ± 2.9 | 10.5 ± 2.6 |
| P18C | BB | 11.7 ± 6.0 | 14.7 ± 5.5 | 14.0 ± 6.5 | 12.9 ± 6.7 | 12.3 ± 7.0 | 10.0 ± 6.8 |
| STAGES | BB | 11.7 ± 4.3 | 14.5 ± 4.2 | 13.9 ± 4.9 | 12.9 ± 4.9 | 12.1 ± 5.0 | 10.1 ± 4.7 |
| MGH | BB | 13.9 ± 2.2 | 16.7 ± 2.5 | 16.3 ± 3.0 | 15.3 ± 3.2 | 14.6 ± 2.8 | 12.2 ± 2.4 |
| MGH | RF | 12.6 ± 2.2 | 15.0 ± 2.9 | 14.7 ± 3.1 | 13.9 ± 3.0 | 13.4 ± 2.6 | 11.1 ± 2.2 |
| **Average** | | 14.1 ± 3.1 | 17.4 ± 3.4 | 16.6 ± 3.8 | 15.5 ± 3.7 | 14.6 ± 3.8 | 12.4 ± 3.4 |

*(b) External datasets*

| Dataset | Source | Overall | $\delta$ (0–4Hz) | $\theta$ (4–8Hz) | $\alpha$ (8–12Hz) | $\sigma$ (12–16Hz) | $\beta$ (12–32Hz) |
|---|---|---|---|---|---|---|---|
| WSC | BB | 14.3 ± 1.9 | 17.0 ± 2.1 | 17.6 ± 2.4 | 16.3 ± 2.4 | 15.5 ± 2.4 | 12.6 ± 2.2 |
| CFS | BB | 13.4 ± 2.1 | 15.7 ± 2.8 | 14.7 ± 3.0 | 14.7 ± 2.8 | 14.0 ± 2.5 | 12.2 ± 2.2 |
| MESA | BB | 13.5 ± 3.3 | 16.4 ± 3.6 | 15.6 ± 4.3 | 15.2 ± 4.3 | 14.3 ± 4.2 | 12.0 ± 3.7 |
| SOF | BB | 13.9 ± 2.2 | 16.4 ± 2.7 | 16.4 ± 3.1 | 16.0 ± 3.1 | 14.9 ± 2.9 | 12.4 ± 2.3 |
| UMASS | RF | 14.2 ± 1.7 | 16.6 ± 1.8 | 16.1 ± 2.2 | 14.7 ± 2.3 | 13.6 ± 2.2 | 12.4 ± 1.8 |
| **Average** | | 13.9 ± 2.5 | 16.6 ± 2.9 | 16.4 ± 3.5 | 15.7 ± 3.3 | 14.8 ± 3.3 | 12.3 ± 2.9 |

ground-truth EEG signals. Of the 14 datasets, 12 collect breathing via wearable belt and one (UMASS) via RF sensing, while MGH includes both paired belt- and RF-based recordings.

Different experiments focus on different datasets depending on availability of labels and input modality, or to match the datasets used to evaluate the baseline. Specifically:

- *Reconstruction from breathing belt* is evaluated on all datasets except UMASS, which lacks belt data.
- *Reconstruction from RF* is evaluated on UMASS and MGH, the only datasets containing RF data.
- *Age estimation from breathing belt* is evaluated on CCSHS, CFS, CHAT, MESA, MGH, MrOS, SHHS, NCHSDB, SOF, and WSC, which have age labels.
- *Sex classification from breathing belt* is evaluated on CCSHS, CFS, MESA, MGH, SHHS, and WSC. Sex classification is evaluated for participants older than 12 years (post-puberty), excluding pediatric datasets, male- or female-only datasets, datasets with no sex labels, and individuals younger than 12 in the rest.
- *Sleep staging from breathing belt* is evaluated on MESA, MGH, SHHS-1, SHHS-2, and WSC. This choice of datasets and the separation of SHHS into two datasets based on visit 1 vs. 2 allows for fair com-

parison with the SOTA in He et al. (2025).
- *RF-based downstream tasks* are evaluated on MGH because it contains paired RF-based and belt-based breathing, allowing for comparing the two modalities.

Additional dataset details are in the Supplemental Material.

### 4.2. Quantitative Reconstruction Results

We evaluate the performance of our cross-physiology model for synthesizing sleep EEG from respiratory signal. Table 1 reports reconstruction performance in terms of MAE. On internal datasets used for training and cross-validation (Table 1a), the model yields high reconstruction quality, achieving an average MAE of $0.068 \pm 0.036$ across EEG frequency bands. Errors are well balanced across bands, suggesting the model captures shared latent structure between respiratory dynamics and EEG activity without overfitting to a particular frequency range.

Crucially, this performance transfers to unseen data. On the external datasets held out from training (Table 1b), the model maintains comparable accuracy, with an average MAE of $0.067 \pm 0.030$. The close agreement between internal and external results demonstrates robustness to domain shift and supports the generalizability of the learned cross-

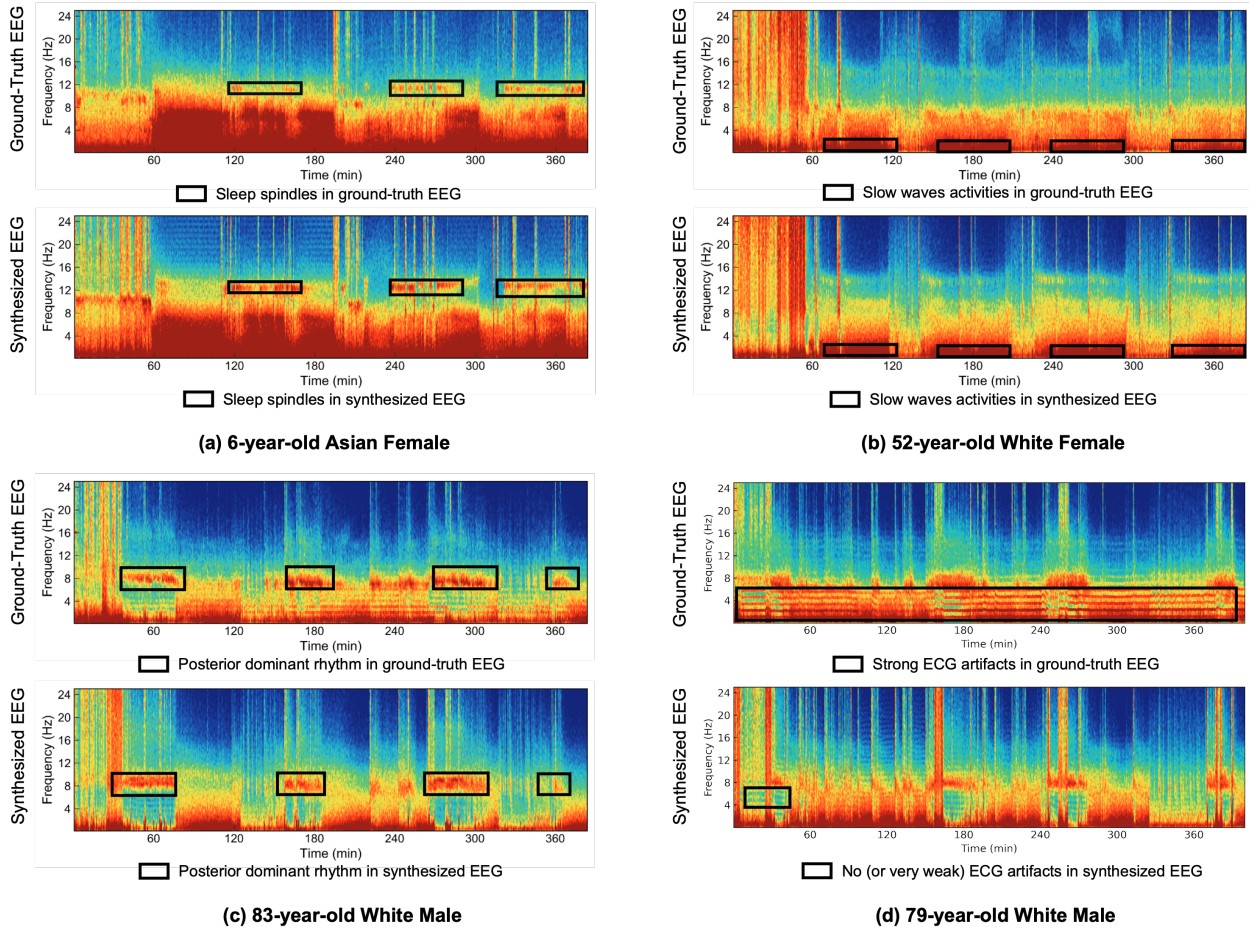

**Figure 4. Visualization of EEG Reconstruction Results.** In each panel, top row shows the ground-truth EEG, and bottom row shows the generated counterpart. The boxes highlight fine features in different EEG bands. These examples underscore the ability of the model to capture and replicate essential EEG features while eliminating some artifacts like the stripes in (d).

modal mapping across datasets and sensing modalities.

Table 2 reports reconstruction performance in terms of SNR. Across datasets, the model achieves a strong average SNR of 14 dB, with values remaining consistent across cohorts and recording conditions. This stability mirrors the MAE results and further supports uniform reconstruction behavior and robust generalization to the external test datasets.

### 4.3. Qualitative Reconstruction Results

We qualitatively assess breathing-to-EEG synthesis by comparing generated EEG spectrograms to their paired ground-truth recordings. Fig. 4 shows representative examples. Across cases, the model preserves salient time-frequency structure and recovers canonical sleep-related patterns, including sleep spindles (Fig. 4a), posterior dominant rhythms (Fig. 4c), and slow-wave activity (Figs. 4a–d). We also observe that synthesized spectrograms are often visually cleaner than the recordings, with reduced artifacts. In partic-

ular, the model suppresses prominent ECG contamination visible as horizontal banding at harmonics of approximately 1 Hz in some recordings, which arises from electrode placement near vasculature (Fig. 4d). Since this contamination is situational EEG sensor-level noise that is statistically independent of the respiratory signal, the model, conditioned purely on breathing dynamics, naturally excludes it, suggesting that the MAE and SNR reported above are conservative: some of the apparent "errors" may reflect benign differences introduced by denoising or artifact removal rather than true reconstruction errors.

### 4.4. Impact of Demographics and Health Conditions

Figs. 5a–c evaluate EEG synthesis accuracy (i.e., MAE) across subgroups defined by age (0–18, 18–40, 40–60, 60–80, 80–100 years), sex, and race (Asian, Black, White, Other). MAE is consistently low with only modest variation. Slightly higher MAE is observed in underrepresented

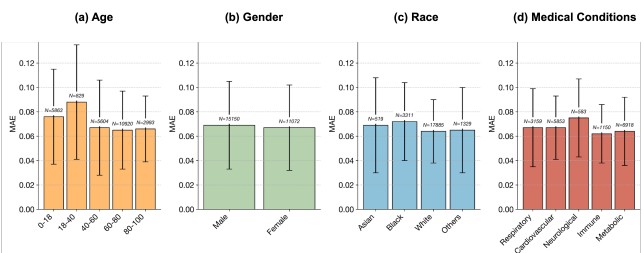

*Figure 5.* **Performance across Demographics and Pre-existing Conditions.** Results are reported as Mean Absolute Error (MAE); lower is better. The figure shows that the MAE stays low across demographics and health conditions.

*Table 3.* **Age prediction performance across datasets.** Results are reported as Mean Absolute Error (MAE) in years ± Standard Deviation across the 4 folds; lower is better.

| Dataset | Breathing | | Synthesized EEG | GT-EEG |
|---|---|---|---|---|
| | CNN | ViT | ViT | ViT |
| CCSHS | 8.10 ± 0.55 | 6.11 ± 0.33 | 2.51 ± 0.65 | 3.97 ± 0.56 |
| CFS | 13.3 ± 0.65 | 10.3 ± 0.55 | 7.34 ± 0.75 | 6.18 ± 0.34 |
| CHAT | 4.07 ± 0.18 | 3.90 ± 0.24 | 2.17 ± 0.39 | 2.51 ± 0.19 |
| MESA | 9.20 ± 0.16 | 8.14 ± 0.26 | 6.30 ± 0.26 | 6.87 ± 0.29 |
| MGH | 12.2 ± 1.05 | 11.6 ± 0.56 | 7.96 ± 0.57 | 7.96 ± 0.67 |
| MROS | 7.40 ± 0.12 | 4.86 ± 0.19 | 5.51 ± 0.72 | 5.07 ± 0.14 |
| NCHSDB | 6.92 ± 0.63 | 6.19 ± 0.37 | 2.97 ± 0.40 | 3.02 ± 0.56 |
| SHHS | 8.95 ± 0.29 | 8.44 ± 0.20 | 6.01 ± 0.06 | 6.21 ± 0.15 |
| SOF | 9.31 ± 0.86 | 9.79 ± 1.18 | 3.85 ± 0.20 | 4.40 ± 0.91 |
| WSC | 8.10 ± 0.43 | 7.28 ± 0.14 | 5.34 ± 0.16 | 5.16 ± 0.23 |
| **Average** | **8.75 ± 0.492** | **7.66 ± 0.402** | **5.00 ± 0.416** | **5.14 ± 0.404** |

sub-groups (ages 18–40; Black and Asian participants) and in children (0–18) who experience greater developmental EEG variability. We expect these gaps to diminish with larger training data from these sub-groups.

We also stratify by major disease categories (cardiovascular, autoimmune, metabolic, neurological, respiratory). As shown in Fig. 5d, the MAE remains low across conditions with marginally lower error for cardiovascular and metabolic disorders and higher error for neurological conditions, likely reflecting smaller sample size.

### 4.5. Performance on Downstream Tasks

The utility of the synthesized EEG was further validated through age estimation, sex prediction, and sleep-staging. We compared these results against baselines trained on raw breathing signals and ground-truth EEG spectrograms.

**Age Estimation.** Table 3 shows age prediction from synthesized EEG closely matches ground truth (MAE 5.14 vs. 5.00 years), significantly outperforming direct prediction from breathing (MAE 7.66 years).

**Sex Prediction.** As shown in Table 4, sex prediction from synthesized EEG nearly matches ground-truth EEG (AUROC=0.814 vs. AUROC=0.819), while breathing-based prediction is weaker (AUROC=0.768).

*Table 4.* **Sex classification performance across datasets.** Results are reported as AUROC (male vs. female) ± Standard Deviation across the 4 folds; higher is better.

| Dataset | Breathing | | Synthesized EEG | GT-EEG |
|---|---|---|---|---|
| | CNN | ViT | ViT | ViT |
| CCSHS | 0.706 ± 0.013 | 0.779 ± 0.040 | 0.773 ± 0.044 | 0.771 ± 0.044 |
| CFS | 0.646 ± 0.079 | 0.768 ± 0.053 | 0.758 ± 0.043 | 0.820 ± 0.043 |
| MESA | 0.749 ± 0.012 | 0.779 ± 0.026 | 0.866 ± 0.013 | 0.851 ± 0.013 |
| MGH | 0.655 ± 0.032 | 0.705 ± 0.033 | 0.749 ± 0.038 | 0.780 ± 0.038 |
| SHHS | 0.778 ± 0.024 | 0.801 ± 0.024 | 0.894 ± 0.025 | 0.864 ± 0.025 |
| WSC | 0.768 ± 0.021 | 0.776 ± 0.010 | 0.845 ± 0.030 | 0.825 ± 0.030 |
| **Average** | **0.717 ± 0.030** | **0.768 ± 0.031** | **0.814 ± 0.032** | **0.819 ± 0.032** |

*Table 5.* **Sleep stage prediction performance across datasets.** Results are reported as night-level Accuracy ± Standard Deviation across the 4 folds; higher is better.

| Dataset | He et al. (2025) | Synthesized EEG | GT-EEG |
|---|---|---|---|
| MESA | 0.799 ± 0.101 | 0.851 ± 0.060 | 0.878 ± 0.077 |
| MGH | 0.810 ± 0.080 | 0.827 ± 0.070 | 0.871 ± 0.063 |
| SHHS-1 | 0.788 ± 0.095 | 0.825 ± 0.069 | 0.877 ± 0.070 |
| SHHS-2 | 0.831 ± 0.063 | 0.841 ± 0.056 | 0.893 ± 0.053 |
| WSC | 0.834 ± 0.084 | 0.850 ± 0.063 | 0.892 ± 0.048 |
| Overall | 0.812 ± 0.085 | 0.839 ± 0.064 | 0.882 ± 0.062 |

**Sleep Stage Classification.** Sleep staging in clinical settings is typically performed from EEG, although recent work has shown that the hypnogram (a sequence of 30-second segments labeled Wake, Light, Deep, or REM) can be inferred directly from breathing. We compare the state-of-the-art breathing-based sleep staging model (He et al., 2025) to our two-stage approach: synthesize EEG from breathing and then predict sleep stages from the synthesized EEG. We also report an upper bound using ground-truth EEG. For a fair comparison, we evaluate on the datasets used in He et al. (2025) and match model capacity across methods.

As shown in Table 5, our breathing → synthesized EEG → staging pipeline improves over the breathing-based baseline (Acc.=0.839 vs. Acc.=0.812) and approaches the performance achieved using ground-truth EEG (Acc.=0.882). To our knowledge, this represents a new state of the art in sleep stage classification result from a breathing modality, narrowing the gap to EEG-based performance. Since sleep stages are heavily imbalanced, we also report macro-F1: synthesized EEG reaches 0.76 versus 0.81 for ground-truth EEG, confirming that the accuracy gains are not an artifact of the dominant stages.

### 4.6. Results for EEG Synthesis from RF Signals

We evaluate whether sleep EEG can be synthesized from contactless RF reflections acquired during sleep. We consider the RF-based cohorts in Tables 1 and 2 (MGH and UMASS). Across RF cohorts, synthesized EEG attains an average MAE of 0.075 and an average SNR of 13.4 dB, which is on par with belt-based breathing. We further test downstream utility on the MGH dataset, which includes paired belt- and RF-based breathing for the same people and

nights, along with age, sex, and sleep stage labels. As shown in Table 6, EEG synthesized from RF breathing supports age estimation, sex prediction, and sleep staging at slightly lower accuracy than EEG synthesized from belt breathing (age MAE 8.8 vs. 7.9 years; sex AUROC 0.70 vs. 0.74; sleep staging Acc. 0.81 vs. 0.82). Overall, these results show for the first time the feasibility of EEG inference from passive wireless sensing without wearable devices.

*Table 6.* **Performance of wireless signals compared to breathing belt (on MGH dataset) across three downstream tasks.**

| Source | BB | RF |
|---|---|---|
| Age (MAE) ↓ | 7.96 ± 0.57 | 8.83 ± 0.75 |
| Sex (AUROC) ↑ | 0.749 ± 0.038 | 0.703 ± 0.049 |
| Sleep Stage (Acc.) ↑ | 0.827 ± 0.070 | 0.814 ± 0.070 |

### 4.7. Disentangling Sleep Stage from Neural Signatures

We verify that our model captures subject-specific neural structure, not just a sleep-stage-conditioned average, through two analyses. First, within each sleep stage, we compute the ratio of between-subject standard deviation in synthesized EEG to that in ground-truth EEG: A stage-average template would yield 0%, while a perfect reconstruction would yield 100%. As shown in Table 7, our model preserves 63–77% of individual variation across stages, with the residual gap reflecting the information bottleneck of discrete tokenization.

*Table 7.* Ratio of between-subject standard deviation in synthesized EEG relative to ground-truth EEG, computed within each sleep stage. A stage-average template would yield 0%.

| Stage | Std. Dev. Ratio (Synth / GT) |
|---|---|
| Wake | 77% |
| Light | 65% |
| Deep | 67% |
| REM | 63% |

Second, we benchmark against an explicit stage-average baseline that replaces every epoch with its sleep-stage mean, and measure the band-wise Pearson correlation of each model's output with ground-truth EEG. Our synthesized EEG improves correlation over the stage-average baseline by 14%, 19%, 28%, 22%, and 8% in the Delta, Theta, Alpha, Sigma, and Beta bands. Since the baseline already encodes all stage-level information, these gains reflect intra-stage structure, which is the fine-grained variation that distinguishes individuals and that distinguishes successive epochs of the same stage within a single night. Together, these results confirm that breathing carries a genuine "fingerprint" of brain activity beyond sleep architecture alone.

## 5. Conclusion

We introduced the task of cross-physiology translation and showed that a waveform-conditional generative framework can decode the intricate, nonlinear coupling between the pulmonary and neurological systems to synthesize high-fidelity sleep EEG directly from breathing dynamics. Evaluated across 14 datasets (33,919 nights; ages 3–102), the method achieves low reconstruction error and generalizes across cohorts and sensing modalities. Notably, we provide the first evidence that contactless RF reflections can support the synthesis of meaningful EEG spectrograms, pointing toward remote, contact-free neurological assessment during sleep. Limitations include a remaining gap to ground-truth EEG and a focus on a single EEG channel (C4–A1), which can be addressed in future work. Overall, our results suggest a scalable path from ubiquitous breathing signals to EEG representations, enabling scalable remote neurophysiological monitoring without head-worn sensors.

## Impact Statement

This work introduces a generative framework capable of translating respiratory dynamics into high-fidelity sleep EEG, utilizing a dataset of over 28,000 individuals. By demonstrating that detailed neurological information can be synthesized from breathing —including via contactless wireless reflections— this research holds the potential to democratize access to sleep EEG. It offers a scalable alternative to traditional polysomnography, potentially benefiting populations for whom wearable EEG is inaccessible or impractical, such as children, the elderly, and patients with sensory processing sensitivities. Realizing this benefit equitably will require attention to demographic coverage: reconstruction error is slightly higher in underrepresented groups in our data, a gap we expect targeted data collection to close.

**Ethical Compliance and Data Governance.** All datasets utilized in this study were collected under strict ethical guidelines. Institutional Review Board (IRB) approval was obtained for all data collection protocols, and informed consent was secured from all participants prior to their inclusion in the source datasets. We strictly adhere to data use agreements that prohibit the re-identification of subjects.

**Privacy and Dual-Use Concerns.** The capability to infer brain activity from contactless wireless signals raises distinct privacy considerations. Unlike wearables, radio-frequency (RF) sensing can theoretically be deployed unobtrusively. While our method requires a learned mapping that generalizes across populations, the possibility of inferring sensitive sleep architecture or biometric attributes (as demonstrated by our age and sex prediction results) from passive environmental signals underscores the need

for robust privacy safeguards. Future deployment of such technology must be governed by frameworks that ensure transparency, explicit user consent, and privacy-by-design principles to prevent unauthorized surveillance.

**Clinical Limitations and Safety.** While our generative model achieves high fidelity (7% MAE) and strong downstream utility, synthesized EEG is a probabilistic reconstruction, not a direct measurement of neural voltage, and a gap to ground-truth EEG remains. The present framework also focuses on a single EEG channel (C4–A1); extending it to multi-channel settings is left to future work. Like all generative foundation models, there is a risk of hallucination, where the model might generate plausible but factually incorrect physiological features, or conversely, fail to capture rare pathological events not well-represented in the respiratory signal. Severe sleep apnea, in particular, can disrupt the respiratory–neurological coupling the model relies on and degrade fidelity, motivating future work on apnea-aware conditioning and on flagging low-confidence segments. Consequently, this technology should be viewed as a screening or support tool rather than a standalone diagnostic replacement. Clinical adoption will require rigorous human-in-the-loop validation to ensure that synthetic approximations are not mistaken for ground-truth physiology in critical care decisions.

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

# A. Supplementary Material

## A.1. Dataset Summary

We conducted a retrospective analysis using a dataset comprising 33,919 nights from 28,394 individuals (mean age 52.6 years, standard deviation 25.1, age range 3–102 years). Of these individuals, 44.7% were female, and 72.8% identified as white. The dataset was constructed by aggregating data from 14 distinct sleep datasets, of which 9 were used for training and cross-validation, while 5 datasets were reserved for external testing (refer to Table. 8).

The MGH dataset comprises polysomnography (PSG) recordings collected at the Massachusetts General Hospital (MGH) Sleep Laboratory between 2019 and 2022. Participants were required to be at least 18 years old, cognitively unimpaired, free from electronic implants, and not pregnant. Upon enrollment and screening, study coordinators installed wireless monitoring devices within the sleep laboratory and ensured connectivity to the clinic's Wi-Fi network. A simplified floor plan of the monitored area was also created to support retrospective assessments of participant positioning. A total of seven sleep technicians independently annotated the dataset, each responsible for one PSG session. Overall, the dataset includes 881 PSG recordings from 881 distinct participants. All study procedures were approved by the Institutional Review Board (IRB) at the Mass General Brigham (IRM no. 2018P000337). The Massachusetts Institute of Technology (MIT) Institutional Review Board ceded review to the Mass General Brigham IRB.

The UMass dataset comprises data collected during an observational home-based study conducted from 2019 to 2021, aimed at investigating wireless signal applications for monitoring movements and vital signs. Participants were required to be adults (18 years or older), have home Wi-Fi access, and be capable of providing informed consent or have consent provided by a legally authorized representative. They also agreed to confidential storage and use of the collected data, acknowledging that anonymized data might be utilized in scientific publications. The study protocol was reviewed and approved by the Massachusetts Institute of Technology Committee on the Use of Humans as Experimental Subjects (COUHES) (IRB no. 1910000024).

The other datasets were obtained from the publicly available National Sleep Research Resource (NSRR) (https://sleepdata.org/datasets).

The datasets encompassed two categories of breathing signal measurements. The first category includes nocturnal breathing data collected using wearable breathing belts (abdominal and thoracic) during polysomnography (PSG) sleep studies. The second category comprises nocturnal breathing signals collected via contactless sensing, utilizing a wireless radio-frequency sensor (Yue et al., 2018) placed in sleep laboratories. This sensor captures breathing patterns by analyzing wireless signal reflections without physical contact or wearable devices. Combining these datasets enabled validation of model performance across different modes of breathing data acquisition and ensured robustness across varied study designs and demographic characteristics.

## A.2. Model and Training Details

### A.2.1. EEG TARGET DESIGN

**Channel selection.** The C4–A1 channel was chosen because it is the standard clinical derivation for sleep EEG, optimally capturing the features foundational to our downstream tasks — slow-wave activity, sleep spindles, and vertex waves. Crucially, the respiratory signal is a single, spatially unresolved measure of global pulmonary effort, which inherently limits recovery of channel-specific spatial information such as anterior-posterior gradients. C4–A1 is well-suited to this input, reflecting global sleep neurophysiology. The framework can extend to multi-channel settings (e.g., a 3D tokenizer over time × frequency × channel), but the respiratory signal's spatial limitation remains a fundamental constraint. We hypothesize that central derivations will perform comparably to C4–A1, while channels sensitive to localized patterns (e.g., occipital alpha) may show larger reconstruction gaps.

**Spectrogram target.** Sleep EEG is clinically defined and interpreted in the frequency domain rather than as a waveform. The AASM scoring rules (Berry et al., 2012) identify sleep stages through frequency-band activity, such as delta power for slow-wave sleep, sigma power for spindles—making the spectrogram the natural representation (Prerau et al., 2017). Beyond staging, spectral features serve as established biomarkers across a range of conditions, including depression (Steiger & Kimura, 2010), PTSD (Denis et al., 2021), schizophrenia (Ferrarelli, 2020), and cognitive health (Sun et al., 2024).

This frequency-domain emphasis also reflects the underlying signal properties. Raw sleep EEG is stochastic, and its clinical content is carried by power spectral density rather than instantaneous phase (Prerau et al., 2017). Predicting waveforms would therefore burden the model with reconstructing exact voltage trajectories whose phase is largely uninformative, while the spectrogram retains precisely the structure that downstream clinical analyses depend on.

### A.2.2. TOKENIZATION

We use a CNN-based VQGAN encoder-decoder and quantizer to compress $256 \times 512$ EEG spectrograms into discrete latent tokens and reconstruct them back. The encoder consists of 5 blocks, each containing 2 residual layers. After each block, the feature map is downsampled via average

*Table 8.* **Summary of datasets used in this study.** External datasets (held out for testing only) are highlighted in grey. Dashes indicate unavailable data.

| Cohort | Usage | Signal | #Part. | #Nights | Female (%) | Age (mean $\pm$ SD) | Asian | Black | White | Other |
|--------|-------|--------|--------|---------|------------|---------------------|-------|-------|-------|-------|
| BWH | Internal | Belts | 4866 | 4866 | 53.2 | $55.1 \pm 17.1$ | 3.5 | 14.1 | 65.7 | 16.7 |
| SHHS (v1 & 2) | Internal | Belts | 5797 | 8444 | 52.4 | $64.5 \pm 11.2$ | 0.0 | 8.9 | 84.6 | 6.5 |
| MROS (v1 & 2) | Internal | Belts | 2906 | 3930 | 0.0 | $77.6 \pm 5.6$ | 2.9 | 3.4 | 91.1 | 2.6 |
| CHAT | Internal | Belts | 1232 | 1639 | 52.2 | $7.0 \pm 1.4$ | 1.2 | 47.6 | 40.3 | 10.9 |
| CCSHS | Internal | Belts | 515 | 515 | 49.5 | $17.7 \pm 0.4$ | 0.2 | 35.9 | 59.6 | 4.3 |
| NCHSDB | Internal | Belts | 3960 | 3960 | 43.1 | $8.8 \pm 6.0$ | 3.2 | 22.5 | 74.3 | 0.0 |
| P18C | Internal | Belts | 1983 | 1983 | 34.9 | $55.0 \pm 14.3$ | – | – | – | – |
| STAGES | Internal | Belts | 1897 | 1897 | 52.2 | $45.9 \pm 15.1$ | 10.2 | 4.2 | 78.4 | 7.2 |
| MGH | Internal | Wireless | 881 | 881 | 43.2 | $54.4 \pm 16.7$ | 4.3 | 7.9 | 78.9 | 8.9 |
| WSC | External | Belts | 1122 | 2569 | 45.9 | $59.8 \pm 8.5$ | 1.1 | 2.0 | 94.8 | 2.1 |
| CFS | External | Belts | 695 | 695 | 54.5 | $41.3 \pm 19.4$ | 0.0 | 55.4 | 41.4 | 3.2 |
| MESA | External | Belts | 2056 | 2056 | 53.6 | $69.4 \pm 9.1$ | 12.2 | 27.8 | 36.1 | 23.9 |
| SOF | External | Belts | 453 | 453 | 100.0 | $82.8 \pm 3.1$ | 0.0 | 8.4 | 91.6 | 0.0 |
| UMASS | External | Wireless | 31 | 31 | – | – | – | – | – | – |
| **Overall** | – | – | 28394 | 33919 | 44.7 | $52.6 \pm 25.1$ | 3.3 | 15.7 | 72.8 | 8.2 |

pooling with scale factors of $(4, 2)$, $(2, 2)$, $(2, 2)$, and $(2, 1)$. The output is then quantized at each spatial location using a codebook with 8192 entries, each with a 32-dimensional embedding. L2 normalization is applied to the latent codes during quantization. The decoder mirrors the encoder structure, consisting of 5 blocks with 2 residual layers each. After every block, the feature map is symmetrically upsampled to progressively reconstruct the original resolution. The model is trained using the Adam optimizer with a learning rate of $4.8 \times 10^{-5}$ for 200 epochs and a batch size of 120.

### A.2.3. MASKED GENERATIVE MODEL

For the masked generative model, we use ViT-Small as the transformer backbone. Both the encoder and decoder consist of 8 transformer blocks with an embedding dimension of 768 and 8 self-attention heads. The model is trained using cross-entropy loss on the masked EEG token positions for 500 epochs using the AdamW optimizer with a peak learning rate of $1.12510^{-4}$, a batch size of 192, and a weight decay of 0.05. We apply linear warm-up learning rate scheduling over the first 40 epochs, followed by cosine learning rate decay.

### A.2.4. DOWNSTREAM MODELS

The age and sex downstream models adopts a ViT-Tiny backbone. They are trained for 33 epochs, while the sleep stage models are trained for 15 epochs. All models employ a 3-epoch learning rate warmup, with a batch learning rate of $1 \times 10^{-3}$ and a weight decay of $1 \times 10^{-2}$. For the contactless monitoring, a single epoch of fine-tuning is applied on wireless data with a learning rate of $5 \times 10^{-4}$ and a weight decay of 1.

### A.3. Ablation Studies

In this section, we present a comprehensive set of ablation studies that empirically justify the key design choices of our framework.

#### A.3.1. DISCRETE SPECTROGRAM TOKENS VS. CONTINUOUS REGRESSION

A central design choice in our framework is to model nocturnal EEG spectrogram synthesis from breathing signals as a classification problem over a finite vocabulary of discrete spectrogram tokens rather than as a continuous regression problem over raw spectrogram values. To isolate the impact of this choice, we implemented a sequence-to-sequence Transformer baseline that takes the same breathing embeddings as input and regresses the EEG spectrogram pixel-by-pixel in continuous space, optimized with an $L_1$ reconstruction loss. All other architectural and training details (backbone, embedding dimension, depth, optimizer, schedule) are held identical to the main model.

As shown in Table 9, the continuous regression baseline increases the average MAE by 40% (from 0.067 to 0.094 on the normalized spectrogram), and produces visibly blurrier outputs that wash out transient EEG features such as sleep spindles and slow-wave bursts. This is consistent with the broader observation in generative modeling that direct regression in high-dimensional, multi-modal output spaces tends to collapse onto the conditional mean, producing over-smoothed samples (Li et al., 2023; Yu et al., 2024). Discrete tokenization, in contrast, constrains the output to a learned codebook of physiologically plausible EEG pattern, reducing mode averaging and preservign sharpness.

*Table 9.* Ablation on the EEG target representation. Mean Absolute Error (MAE) is computed on normalized EEG spectrograms. Discrete tokenization substantially outperforms pixel-wise regression.

| EEG Target Representation | MAE $\downarrow$ |
|---|---|
| Continuous regression (seq2seq Transformer) | 0.094 |
| Discrete tokens (**ours**) | **0.067** |

### A.3.2. TOKENIZER DESIGN

We next ablate the four main hyperparameters of the VQ-GAN tokenizer: codebook size, token patch resolution, discriminator architecture, and learning-rate schedule. Each ablation modifies a single factor while holding all others fixed at the values used in the main model (codebook of 8,192 entries with dimension 32; patch size 32×8, i.e. $\sim 4\,\mathrm{Hz} \times 4\,\mathrm{min}$; StyleGAN-style discriminator with $R_1$ regularization; cosine learning-rate decay).

**Codebook size.** Reducing the codebook from 8,192 to 1,024 entries increases $L_1$ reconstruction error by 5%. The modest degradation indicates that the tokenizer is robust to this choice, but a larger codebook provides a richer vocabulary of EEG patterns and slightly improves fidelity.

**Token patch resolution.** Our default patch of 32×8 corresponds to approximately $4\,\mathrm{Hz} \times 4\,\mathrm{min}$, deliberately chosen so that each token row isolates a canonical EEG band ($\delta$, $\theta$, $\alpha$, $\sigma$) and each token column aggregates enough temporal context to capture a stable meso-scale sleep state. Replacing this with a 256×4 patch, which collapses the entire frequency axis into a single token, increases $L_1$ error by 12.7%. Finer resolutions (e.g. 16×4, corresponding to $\sim 2\,\mathrm{Hz} \times 2\,\mathrm{min}$) were computationally prohibitive: halving both patch dimensions quadruples the token sequence length and increases self-attention cost by roughly 16×.

**Discriminator architecture.** We compared a PatchGAN discriminator (a common default in VQGAN implementations) against a StyleGAN-style discriminator with $R_1$ gradient penalty. The StyleGAN variant reduces $L_1$ error by 10% and noticeably stabilizes training; we therefore adopt it in the main model.

**Learning-rate schedule.** Cosine annealing and a constant learning rate produce no statistically significant difference in $L_1$ error. We use cosine annealing in the main model for consistency with the rest of the training pipeline.

**Codebook utilization.** A common failure mode of VQ-based models is *codebook collapse*, in which only a small fraction of codes are used. To rule this out, we measured codebook utilization at convergence under two regimes: (i)

when the tokenizer encodes ground-truth EEG spectrograms, and (ii) when the translation generator emits tokens conditioned on breathing. Under (i), 100% of codebook entries are used, facilitated by the $L_2$ normalization of latent codes during quantization (Appendix A.2.2). Under (ii), 79.8% of codes are used, confirming that the generator draws on a diverse vocabulary rather than collapsing onto a small subset of patterns.

### A.3.3. GENERATIVE TRANSLATION OBJECTIVE

In addition to the masked generative modeling approach, as described in Sec. 3.2, we also conducted ablation experiments using autoregressive (AR) generation to validate our choice of translation objective. We tested two autoregressive ordering strategies:

(1) **Sequential Autoregressive**: We flattened the 2D EEG token grid into a 1D sequence by ordering tokens from low to high frequency within each time step, and then advancing sequentially from left to right across time. This strictly causal formulation resulted in a 21.5% increase in Mean Absolute Error (MAE) compared to the masked model.

(2) **Column-wise Parallel Autoregressive**: We grouped the vertical column of frequency tokens at each time step into a single unit and performed autoregressive prediction over these temporal chunks. This approach, while capturing instantaneous spectral structure better than the fully sequential baseline, still resulted in a 14.8% increase in MAE.

We attribute the superiority of the masked approach to the inherent characteristics of the physiological translation task. Unlike natural language generation, synthesizing a sleep spectrogram from breathing is a *global* translation problem, where the structural consistency across the entire night is essential. Autoregressive models are restricted to conditioning on past EEG tokens, making them vulnerable to error accumulation and unable to utilize future contexts. In contrast, the masked generative objective enables bidirectional attention, allowing the model to infer physiological states from both preceding and succeeding context to produce a holistically coherent spectrogram.

### A.3.4. MASKING SCHEDULE

During training, we sample the masking ratio $\gamma$ from a truncated Gaussian ($\mu = 0.55$, clipped to $[0.5, 1.0]$), following Li et al. (2023). This variable schedule exposes the model to a spectrum of difficulty levels: high masking ratios encourage generation from breathing alone, while lower ratios expose the model to partial EEG context and encourage it to learn the co-occurrence structure both among EEG tokens and between EEG and breathing tokens.

To isolate the contribution of this design, we trained an otherwise identical model with a *fixed* masking ratio of 100%

throughout training, matching the inference-time condition. Fixed full masking increases MAE by $4.7\%$. This indicates that exposing the model to partial EEG context during training is beneficial even though all EEG tokens are masked at inference time, presumably because the additional contextual learning regularizes the cross-modal mapping.

### A.3.5. BREATHING INPUT REPRESENTATION

Finally, we examine whether our downstream gains are attributable to genuine cross-modal translation from breathing to EEG, or merely to the use of a spectrogram-based input representation. We trained the same downstream ViT/Ti backbone directly on a *breathing spectrogram* (rather than the synthesized EEG spectrogram), with all other training details held fixed.

*Table 10.* Ablation on the breathing input representation, evaluated on downstream tasks. Training the same downstream model directly on breathing spectrograms (bypassing EEG translation) performs substantially worse than training on our synthesized EEG, confirming that the gains come from cross-modal translation rather than the spectrogram representation alone.

| Downstream input | Age MAE (yrs) ↓ | Sex AUROC ↑ |
|---|---|---|
| Breathing spectrogram | 14.0 | 0.70 |
| Synthesized EEG (**ours**) | **5.0** | **0.81** |

As shown in Table 10, downstream performance from a breathing spectrogram is substantially worse than from our synthesized EEG on both age estimation (MAE 14.0 vs. 5.0 years) and sex classification (AUROC 0.70 vs. 0.81). This rules out the hypothesis that the improvements over the raw-breathing baselines simply reflect the change of input representation: the gains stem specifically from the translation of breathing into EEG.

This result is consistent with our earlier finding (Sec. 3.1) that heavier breathing encoders and time–frequency conversions on the input side *degrade* translation performance. Together, the two results support the design choice of preserving the breathing signal as a raw waveform on the input side while constraining the EEG output through discrete spectrogram tokenization—the asymmetric representation strategy at the core of our framework.

### A.4. Additional Clinical Downstream Tasks

To further probe whether the synthesized EEG preserves clinically meaningful neural information, we evaluated two additional downstream tasks that are classically assessed using sleep EEG: narcolepsy detection and Alzheimer's disease detection. For each task, we compared models trained on synthesized EEG against the same ground-truth EEG and raw breathing baselines used throughout the paper, with all architectural and training details held fixed.

*Table 11.* Performance on additional clinical downstream tasks. Results are reported as AUROC; higher is better. Synthesized EEG matches or closely approaches ground-truth EEG and substantially outperforms direct prediction from breathing.

| Task | Breathing | Synthesized EEG | GT-EEG |
|---|---|---|---|
| Narcolepsy | 0.67 | 0.76 | 0.76 |
| Alzheimer | 0.72 | 0.75 | 0.76 |

As shown in Table 11, synthesized EEG matches ground-truth EEG on narcolepsy detection (AUROC 0.76 vs. 0.76) and approaches it on Alzheimer's detection (AUROC 0.75 vs. 0.76), while substantially outperforming the breathing baseline on both tasks. These results provide preliminary evidence that the translation captures condition-specific neural signatures rather than only coarse sleep architecture. A deeper investigation of clinical utility, including multi-site validation, comparison against established screening tools, and additional neurological conditions, is left to future work.

### A.5. More Examples of EEG reconstruction

Additional examples of EEG reconstructions from breathing are visualized in Fig. 6. In each group, the top row display the ground-truth EEG, while the bottom row show the generated counterpart.

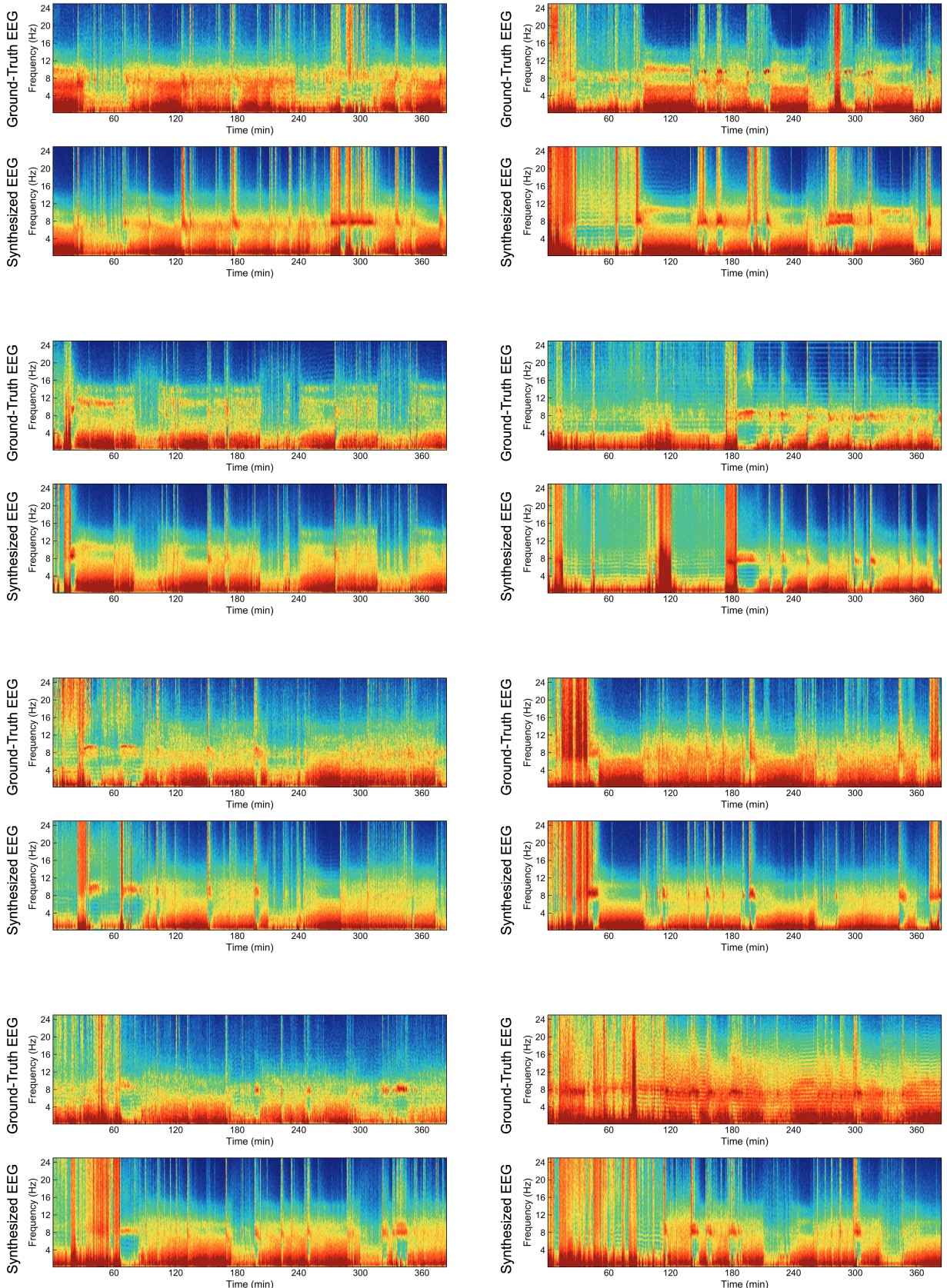

*Figure 6.* More examples of EEG reconstruction.

