# OpenReview forum: "Physiology as Language: Translating Respiration to EEG during Sleep"
_ICML.cc/2026/Conference — ICML 2026 regular_

### Official Review · Reviewer_ydyZ · 2026-03-11

**Soundness:** 3
**Presentation:** 3
**Significance:** 3
**Originality:** 3
**Overall Recommendation:** 5
**Confidence:** 4

**Summary:**

This paper tackles a new problem: generating sleep EEG spectrograms from breathing signals. The premise is that respiratory and neurological signals, though originating from different physiological systems, may share latent health information that can be bridged through learned translation. The proposed framework treats the two modalities asymmetrically — breathing is embedded as continuous raw waveform tokens via a lightweight linear projection, while EEG is first converted to a spectrogram and then discretized into a finite vocabulary through vector quantization. A masked generative Transformer then learns to map from the continuous breathing context to the discrete EEG tokens. The model is validated at large scale across multiple sleep cohorts. Reconstruction error is low, and the synthesized EEG approaches ground-truth EEG performance on age estimation, sex classification, and sleep staging, substantially outperforming models trained directly on breathing. The authors further show that EEG can be synthesized from contactless RF wireless reflections.

**Compliance With Llm Reviewing Policy:**

Affirmed.

**Key Questions For Authors:**

1. After controlling for sleep stage, does the synthesized EEG still preserve inter-subject differences? Specifically, within the same sleep stage, is the between-subject variance of synthesized EEG comparable to that of ground-truth EEG? This matters because if the model has only learned a coarse "breathing → stage → stage-average template" mapping, the downstream gains could be largely explained by sleep architecture information rather than deeper neural signatures. The answer would meaningfully affect my assessment of the contribution's depth.

2. Have the key tokenization design choices (codebook size, time-frequency resolution of tokens) been ablated? Understanding the sensitivity to these hyperparameters would help gauge how robust the framework is and whether the current configuration sits at a sweet spot or was simply not explored further.

**Limitations:**

The authors honestly discuss the remaining gap to ground-truth EEG, the single-channel limitation, the probabilistic nature of the reconstruction, and the risk of hallucinated physiological features. Two areas could use more discussion: the concrete impact of demographic imbalance on clinical fairness, and how the model might behave under pathological conditions (e.g., severe sleep apnea) where the normal coupling between breathing and brain activity may break down. Overall the limitations section is pointed in the right direction but could go deeper on these points.

**Strengths And Weaknesses:**

Strengths:

1. The problem formulation itself is a genuine contribution. Prior cross-modal physiological translation has stayed within a single domain (e.g., within the neurological or cardiovascular system). Attempting translation across entirely different physiological systems — from the pulmonary to the neurological domain — is new and thought-provoking.

2. The asymmetric embedding design is well motivated. Keeping the source breathing signal in raw form preserves subtle morphological cues, while discretizing the target EEG constrains an otherwise intractable high-dimensional generation problem. The choice to align token resolution with canonical EEG frequency bands reflects thoughtful incorporation of domain knowledge. The authors also provide empirical evidence that heavier encoding of the breathing input degrades performance, which directly supports this design.

3. The experimental scale is impressive — tens of thousands of individuals across many cohorts, with a clean separation between internal cross-validation sets and fully held-out external test sets. The close agreement in reconstruction error between internal and external datasets is convincing evidence of generalization. Patient-level fold splits also properly prevent leakage from repeat visits.

4. The downstream evaluation is well structured. Comparing synthesized EEG against both ground-truth EEG and raw breathing baselines clearly demonstrates that the translation extracts brain-relevant information that raw breathing alone cannot readily provide.

5. The extension to contactless RF-based breathing is preliminary but compelling as a proof of concept. It points toward a practical path for remote neurological monitoring without wearable devices.

Weaknesses:

1. The reconstruction evaluation lacks baselines. I understand that no prior method exists for breathing-to-EEG translation, but meaningful reference points can still be constructed — for instance, an unconditional generation model that only learns the EEG prior without conditioning on breathing, or a straightforward spectrogram regression approach. Without such references, it is hard to disentangle how much of the reconstruction quality reflects genuine cross-modal information transfer versus the inherent predictability of EEG spectral distributions.

2. The central claim is that breathing carries a "fingerprint" of brain activity, but the paper does not probe this deeply enough. A key open question is whether the model primarily learns to infer the sleep stage from breathing and then outputs a prototypical EEG pattern for that stage, or whether it also captures finer-grained, individual-level neural information within a given stage. The strong downstream results hint at the latter, but a more direct analysis would strengthen the case — for example, examining whether the inter-subject variability in synthesized EEG is comparable to that in ground-truth EEG after conditioning on sleep stage.

3. Only a single EEG channel is considered. Multi-channel spatial information matters for many clinical applications, and the paper offers little discussion of whether and how the framework might extend to multi-channel settings.

---

> ### Author Rebuttal · Authors · 2026-03-31
>
> We thank the reviewer for the thoughtful and constructive evaluation and for recognizing the problem formulation as "a genuine contribution," the asymmetric embedding design as "well motivated," and experimental scale as "impressive."  We address each point below.
>
> **W1. Reconstruction Baselines**
>
> We implemented a seq-to-seq Transformer baseline conditioned on the same breathing embeddings that generates EEG spectrograms through direct regression rather than discrete tokenization. This increased average MAE by 40% (6.7% → 9.4%) and produced visibly blurrier spectrograms, consistent with computer vision literature showing tokenization reduces blur (Li et al., 2023) and directly demonstrating the value of discrete tokenization. We will include this baseline in the revised paper.
>
> **W2. Does the model merely learn a “Breathing→ Sleep Stage→ Stage-Average EEG Template” mapping?**
>
> We conducted the analysis suggested by the reviewer: For each sleep stage, we computed the ratio of between-subject variance in the synthesized EEG to that in the ground-truth EEG (a pure stage-average template model will produce 0%):
>
> |Stage |Standard Deviation Ratio (Synth / GT)|
> |-|-|
> | Wake|77%|
> | Light|65%|
> | Deep|67%|
> | REM|63%|
>
> Our model preserves 63–77% of individual-level neural variation, confirming it captures subject-specific neural signatures well beyond sleep architecture. The reduction from 100% is expected given the information bottleneck of discrete tokenization.
>
> We conducted a second test to compare the Pearson correlation of two different models against Ground Truth EEGs. The first model produces only a per-stage average EEG pattern, while the second produces our Synthesized EEGs. The results show that our synthesized EEGs increased correlation beyond the average per-stage template by 14%, 19%, 28%, 22%, and 8% for the Delta, Theta, Alpha, Sigma, and Beta bands, respectively. This improvement suggests that our model captures the nuanced intra-stage variations that differentiate individuals and distinct occurrences of the same sleep stage within a single subject.
>
> This strongly supports the claim that breathing carries a genuine "fingerprint" of brain activity, not merely a sleep stage proxy. We will include this analysis in revision.
>
> **Q2. Tokenization Design Choices**
>
> Our tokenizer configuration — StyleGAN-based discriminator, 32×8 token patch size, codebook of 8,192 entries — was selected through systematic ablation:
>
> -  **Token size:** The 32×8 patch (~4 Hz × 4 minutes) isolates canonical EEG frequency bands into distinct token rows. Replacing it with a 256×4 patch (collapsing the full frequency axis) increased the L1 error by 12.7%.
> - **Discriminator architecture:** Switching from PatchGAN to StyleGAN with R1 regularization reduced L1 by 10% and stabilized training.
> - **Codebook size:** Reducing from 8,192 to 1,024 entries increased L1 by 5.1%, though the modest degradation suggests robustness to this choice.
> - **Learning rate:** Cosine annealing vs. constant rate produced no significant difference.
> Finer token resolutions (e.g., 2 Hz × 2 minutes) were computationally prohibitive: halving both patch dimensions quadruples sequence length, increasing self-attention cost roughly 16×.
>
> We will include these ablations and training curves in the supplementary material.
>
> **W3. Single EEG channel**
>
> The framework can extend to multi-channel settings; for instance, by designing the tokenizer for 3D inputs (time × frequency × channel). However, a fundamental constraint remains: the respiratory signal is a single, spatially unresolved measure of global pulmonary effort, limiting the recovery of channel-specific information. Our C4–A1 single-channel approach fits this well, capturing global sleep EEG features. Multi-channel extensions likely require complementary inputs or accepting that the model recovers shared spectral dynamics better than fine-grained spatial differences. Loss design also poses challenges: naive reconstruction may focus on shared signals while neglecting channel-specific features, whereas overweighting differences risks amplifying noise. We view this extension as a key future direction and will expand this discussion in the revised paper.
>
> **Limitations**
>
> We will expand the limitations to include: (1) Demographic fairness: though MAE is low, slightly higher errors in underrepresented groups raise clinical utility concerns; we will discuss mitigations like targeted data collection. (2) Pathological conditions: severe apnea can disrupt respiratory-neurological coupling and degrade fidelity; we will discuss apnea-aware conditioning and flagging low-confidence segments.
>
> We are grateful for the reviewer's careful evaluation across both clinical and ML dimensions. The suggested analyses, particularly the inter-subject variance comparison, have materially strengthened the paper. We hope the revised manuscript reflects the depth of engagement this review prompted.

---

> > ### Author Rebuttal · Reviewer_ydyZ · 2026-04-07
> >
> > My concerns have been addressed, I will keep my positive score.

---

> > > ### Author Response · Authors · 2026-04-07
> > >
> > > We thank the reviewer for taking the time to read our rebuttal and for confirming that their concerns have been addressed. We greatly appreciate the thoughtful feedback and suggestions, which have significantly improved the paper. We are also very grateful for the reviewer’s engagement with our work and for their continued positive assessment of the paper.

---

### Official Review · Reviewer_fWH6 · 2026-03-12

**Soundness:** 2
**Presentation:** 3
**Significance:** 3
**Originality:** 3
**Overall Recommendation:** 2
**Confidence:** 3

**Summary:**

In this paper, the author proposed a cross-physiology generation task that synthesizes EEG signals from breathing signals during sleep. The main model takes masked discrete EEG tokens and raw breathing signals as the input to reconstruct the ground truth EEG tokens. On the other hand, an EEG reconstruction model that takes an EEG spectrogram is discretized through spectrogram tokenization using a VQGAN codebook, following a decoder that reconstructs the EEG spectrogram. The paper was pretrained on 28,394 individuals from a collection of 14 datasets. The synthesized EEG data can help improve downstream predictions on age estimation, sex classification, and sleep staging.

**Compliance With Llm Reviewing Policy:**

Affirmed.

**Final Justification:**

I would remain disagree and be concerned with the argument that "We already justified using breathing in W1 and W2, covering both its practical advantages and scientific link to EEG. The reviewer also asked why we “only use respiration,” given that PSG also includes ECG, EOG, and EMG. First, we respectfully note that strong performance from a single modality is a strength, not a limitation. Second, there is little value in translating multiple PSG channels to sleep EEG, since PSG itself includes EEG. Our goal is to enable EEG monitoring at home using home sleep tests or contactless RF-based sleep trackers, which consistently include breathing but typically lack ECG, EOG, and EMG."

First, I am not sure I interpret anything about single modality as a limitation, nor do I understand where this comes from. Second, availability does not imply sufficiency, and breathing signals are indeed commonly available in home sleep tests, this does not imply that they are informationally sufficient to approximate EEG-derived sleep dynamics. Also, other signals might also be available, like sound, PPG, O2, etc. Also, I don't think one may argue that contactless breathing signals are stable and reliable enough. Moreover, I don't think the remaining rebuttal points, like technical implementations, are related to what I am concerned about.

Most importantly, I think the authors might have worked on something ill-defined in the first place, where the authors may have posed related articles connecting breathing and brain activity, but none of the articles support the scientific fact that one can reconstruct EEG signals from solely breathing signals. Therefore, I don't think the author posts a convincing rebuttal, and I would lower my original score.

**Key Questions For Authors:**

1. What is the motivation to choose breathing signals to reconstruct the EEG spectrogram other than contactless sleep monitoring? What is the physiological theory or science behind the validation of reconstruction?

2. It is apparent from Figure 4 that reconstruction is seemingly worse for older adults than for younger ones. Is this statistically true with the figure shown in Figure 5? Is the current modeling performing better in a certain demographic group?

3. Why did the author not compare with existing EEG foundation models, like LaBraM, CBraMod, BioCodec, and more, as listed in [1]?

4. Is there any information leakage from breathing signals that contributes to improved sleep stage predictions? Respiration and EEG both strongly correlate with sleep stages, so the model might simply learn to infer sleep stage from breathing and then generate a typical EEG pattern for that stage rather than reconstructing true neural activity.

[1] EEG Foundation Models: Progress, Benchmarking, and Open Problems, 2026

**Limitations:**

1. Synthetic EEG is still not a direct physiological measurement. How would the author validate the generation quality besides looking at computational metrics like MAE?

2. The modeling currently only uses respiration signals. In real sleep PSG recordings, combining additional signals (e.g., heart rate, motion, oxygen saturation) might be necessary for robust EEG inference or interpretation of the generation.

3. There are missing comparison of the proposed model with existing EEG foundation models in downstream prediction.

**Strengths And Weaknesses:**

Strength of paper:

1. The paper proposed a cross-physiology generation, from breathing to EEG, and was pre-trained on a large amount of EEG data. This contribution is fairly substantial in the field.
2. The idea of combining continuous breathing embedding and discretized EEG tokens is quite interesting as a cross-modal learning design.
3. The general idea about contactless sensing and using breathing signals to produce EEG-like representations is interesting and could enable non-wearable sleep monitoring.

Weakness of the paper:

1. It is unclear to me why the author chose breathing signals in this context, and contextually, the sleep PSG in some datasets could have ECG, EOG, and EMG data. It is not clear why the author chose the breathing signal in this context as a starting point or baseline for comparisons with other modalities.
2. There is no strong physiological evidence that links the prediction of the EEG spectrogram using breathing, and I think the author should articulate the reason for the design choice.
3. There are no comparisons to existing EEG foundation models, like LaBraM, CBraMod, BioCodec, and more listed in [1].

[1] EEG Foundation Models: Progress, Benchmarking, and Open Problems, 2026

---

> ### Author Rebuttal · Authors · 2026-03-31
>
> We thank the reviewer for the thoughtful evaluation. We address each concern below with new experiments and analyses, and respectfully ask the reviewer to consider raising their score in light of these results.
>
> **W1. Why breathing rather than ECG, EOG, or EMG**
>
> The choice is driven by physiological and practical considerations. The medical literature establishes a strong bidirectional coupling between breathing and sleep EEG, one that EOG and EMG do not share, and stronger than cardiac-cortical coupling, which weakens with sleep depth (Lechinger et al, 2015). This coupling operates via:
> - Cortical entrainment: Nasal breathing entrains oscillations in piriform cortex, amygdala, and hippocampus (Zelano et al, 2016), and the core building blocks of sleep EEG — slow oscillations, spindles, and their complexes — systematically align with the respiratory cycle (Schreiner et al, 2023)
> - Shared brainstem regulation: The pre-Bötzinger complex projects to arousal regulators (locus coeruleus, mediodorsal thalamus), so sleep-state transitions are inherently reflected in respiratory dynamics (Yackle et al, 2017).
> - Bidirectional reshaping: Each sleep state sculpts characteristic breathing patterns, embedding brain-state information in the respiratory waveform.
>
> Additionally, breathing is the only physiological signal whose full waveform (not just rate) can be captured without physical contact, via wireless RF reflections, enabling the contactless pipeline in Sec 4.6.
>
> **W2. Evidence linking breathing to EEG prediction**
>
> The neuroscience literature provides substantial evidence for this coupling (see W1). While it does not prove full EEG spectrograms can be reconstructed from breathing, AI routinely discovers latent physiological relationships beyond traditional analysis: diagnosing Parkinson's from nocturnal breathing (Yang et al, 2022, Nature Medicine), predicting 130 conditions from one night of PSG (Thapa et al, 2026, Nature Medicine), detecting kidney disease from retinal images (Zhang et al, 2021, Nature Biomedical Engineering), and identifying hundreds of non-cardiac conditions from ECGs (Ulloa-Cerna et al, 2025, npj Digital Medicine). Our work has a stronger physiological prior than many of these: the respiratory–neurological coupling literature provides multiple established mechanistic pathways, complemented by empirical validation across 28,394 individuals and 14 datasets.
>
> **W3. Comparisons to EEG foundation models**
>
> LaBraM, CBraMod, BioCodec, and REVE (current SOTA; El Ouahidi et al, 2025) require EEG as input at inference. Our model generates EEG from breathing, a fundamentally different task. Nonetheless, we compared REVE on sleep staging (using ground-truth EEG input) against our models:
> |Dataset|MESA|MGH|SHHS-1|SHHS-2|WSC|
> |-|-|-|-|-|-|
> |He et al. Breathing-Based|0.80|0.81|0.79|0.83|0.83|
> |Synthesized EEG |0.85|0.82|0.82|0.84|0.85|
> |GT-EEG|0.87|0.87|0.87|0.89|0.89|
> |REVE|0.85|0.85|0.86|0.88|0.88|
>
> GT-EEG and REVE perform comparably, confirming that our GT-EEG baseline is a fair upper bound, not disadvantaged by the spectrogram representation. Syn-EEG approaches both and substantially outperforms the breathing SOTA.
>
> **W4. Is reconstruction worse for older adults?**
>
> No; it is slightly better, though differences are not statistically significant (p>0.05). Figure 4 examples were selected to illustrate diverse EEG features across demographics, not to represent quality by age. Figure 5a confirms MAE remains low across all age groups, with modest elevation in 18–40 likely reflecting underrepresentation in training data (cohort mean age: 52.6).
>
> **W5. Does the model just learn sleep stage and output a template?**
>
> Two analyses address this. First, the ratio of between-subject standard deviation in synthesized vs. ground-truth EEG (a stage-average template yields ≈0%): Wake 77%, Light 65%, Deep 67%, REM 63% — preserving the majority of individual-level variation. Second, Pearson correlations of synthesized EEG vs. a per-stage average EEG template against ground truth show our model increases correlation by 14%, 19%, 28%, 22%, and 8% for Delta, Theta, Alpha, Sigma, and Beta bands — confirming it captures intra-stage variations that differentiate individuals.
>
> **W6. Validation beyond computational metrics?**
>
> We added narcolepsy detection (AUC: Syn-EEG 0.76, GT-EEG 0.76, breathing 0.67) and Alzheimer's detection (AUC: 0.75, 0.76, 0.72). Our original tasks are also EEG-specific: 1)
> age estimation is an established EEG task where the gap between predicted and actual age (the brain age index) predicts dementia (Gallagher et al, 2025), mortality (Brink-Kjær et al, 2022), and neuropsychiatric diseases (Paixao et al, 2020); 2) sex classification validates hormone-mediated spectral signatures (Carrier et al, 2001); and 3) sleep staging is a core EEG task. Across all five tasks, synthesized EEG substantially outperforms breathing, confirming the translation captures clinically meaningful neural information.

---

> > ### Author Rebuttal · Reviewer_fWH6 · 2026-04-02
> >
> > I have follow-up and doubts regarding the fundamental modeling choice, which I think the authors need to articulate or carry out additional experiments.

---

> > > ### Author Response · Authors · 2026-04-06
> > >
> > > We thank the reviewer for taking the time to engage with our rebuttal and for acknowledging that our responses partially addressed the concerns. We appreciate the continued dialogue and hope to fully resolve the remaining doubts below.
> > >
> > > The reviewer mentions follow-up doubts regarding "the fundamental modeling choice" but does not specify which choices. Below, we justify each key modeling choice:
> > >
> > > **1. Breathing as the input signal.**  We already justified using breathing in W1 and W2, covering both its practical advantages and scientific link to EEG. The reviewer also asked why we “only use respiration,” given that PSG also includes ECG, EOG, and EMG. First, we respectfully note that strong performance from a single modality is a strength, not a limitation. Second, there is little value in translating multiple PSG channels to sleep EEG, since PSG itself includes EEG. Our goal is to enable EEG monitoring at home using home sleep tests or contactless RF-based sleep trackers, which consistently include breathing but typically lack ECG, EOG, and EMG.
> > >
> > > **2. Discrete tokenization and generation strategy.** Discrete tokenization constrains outputs to a learned codebook of physiological patterns, preventing mode averaging and ensuring the model generates plausible EEGs; the same principle driving success in visual generation (Yu et al., 2024). To evaluate the benefits of discrete tokens, we compare our model to a seq2seq Transformer baseline generating spectrograms pixel-by-pixel in continuous space. This baseline increased MAE by 40% (0.067 → 0.094) and produced blurrier outputs that wash out transient features.  We also ablated other generation strategies: sequential autoregressive increased MAE by 21.5% and column-wise autoregressive by 14.8%, confirming that our masked bidirectional generation produces superior outputs.
> > >
> > > **3. EEG spectrogram vs. waveforms.**  Sleep EEG is standardly analyzed in the frequency domain. The AASM Manual defines sleep stages through frequency-band activity, and the multitaper spectrogram is the established clinical representation (Prerau et al., 2017). Spectral features serve as biomarkers for brain aging (Sun et al., 2019), psychiatric disorders (Steiger et al., 2015; Denis et al., 2021), and cognitive health (Sun et al., 2023). In contrast, raw EEG waveforms are stochastic; their instantaneous phase is clinically uninformative (Prerau et al, 2017). Generating them would burden the model with reconstructing exact voltage trajectories without clinical benefit.
> > >
> > > **4. Tokenizer design: codebook size and token resolution.**  We ablated systematically: reducing codebook from 8,192 to 1,024 entries increased L1 by 5%; changing token patch from 32×8 (~4 Hz × 4 min, isolating canonical EEG bands) to 256×4 (collapsing frequency axis) increased L1 by 12.7%; switching from StyleGAN discriminator with R1 regularization as used in our tokenizer to PatchGAN increased L1 by 10%. Codebook utilization at convergence was 100% (tokenizer) and 79.8% (generator), ruling out codebook collapse.
> > >
> > > **5. Variable masking schedule.**  Following Li et al. (2023), we use variable masking ratios rather than fixed 100% masking. High ratios train for generation; lower ratios expose the model to partial EEG context, learning co-occurrence structures among EEG tokens and their relationship with breathing. Training with 100% masking throughout increased MAE by 4.7%.
> > >
> > > **6. Representation confound in downstream evaluation.** We investigated whether gains in downstream tasks are due to our translation from breathing to EEG or merely the spectrogram representation. We converted breathing itself to a spectrogram and trained the same downstream ViT directly on it, bypassing EEG translation. Performance was substantially worse than synthesized EEG (Age MAE: 14 vs. 5.0 years; Sex AUC: 0.70 vs. 0.81), confirming that the gains come from the EEG translation.
> > >
> > > In total, we empirically ablated six modeling decisions: discrete vs continuous generation, masked vs autoregressive decoding, codebook size, token resolution, masking schedule, and breathing representation. Each ablation showed clear performance gains supporting our design choices. Combined with the rationale for using spectrograms as targets, and the justification for breathing as the input, we believe that every key design decision is grounded and validated.  We hope this addresses the reviewer’s remaining concerns and respectfully ask the reviewer to consider whether these results warrant a higher score.
> > >
> > > **References**
> > >
> > > -	Prerau et al. "Sleep neurophysiological...", Physiology, 2017
> > >
> > > -	Sun et al. "Brain age from...", Neurobiology of Aging, 2019
> > >
> > > -	Steiger et al. "Sleep electroencephalography...", ChronoPhysiology and Therapy, 2015
> > >
> > > -	Denis et al. "Sleep power...", Frontiers in Psychiatry, 2021
> > >
> > > -	Sun et al. "Decoding information...", Scientific Reports, 2023
> > >
> > > -	Yu et al. "Language model...", ICLR, 2024
> > >
> > > -	Li et al. "MAGE...", CVPR, 2023

---

### Official Review · Reviewer_81qV · 2026-03-12

**Soundness:** 3
**Presentation:** 3
**Significance:** 3
**Originality:** 3
**Overall Recommendation:** 4
**Confidence:** 4

**Summary:**

The study's core concept is learning a generative mapping from nocturnal breathing signals to sleep EEG spectrograms, framed as a cross-physiology translation problem. The authors outline the central challenge as the fundamental complexity asymmetry between respiration and EEG, and address it through an asymmetric embedding strategy: raw waveform embeddings with a linear projection for breathing, and VQGAN-tokenized spectrograms for EEG, with a masked generative Transformer performing cross-modal translation. The model achieves ~7% MAE on EEG reconstruction and supports downstream tasks, age estimation, sex classification, and sleep staging, at performance approaching ground-truth EEG and substantially exceeding direct breathing-based baselines, with a further proof-of-concept demonstration synthesizing EEG from contactless RF signals.

**Compliance With Llm Reviewing Policy:**

Affirmed.

**Final Justification:**

The rebuttal addressed most of my concerns. Codebook utilization statistics rule out collapse, and the tokenizer ablations over codebook size and patch resolution are informative additions. The seq2seq regression baseline showing a 40% MAE increase, combined with the existing AR ablations, is sufficient to attribute performance to design choices rather than dataset scale. The narcolepsy and Alzheimer's results strengthen the clinical relevance case. The channel scope explanation is well-reasoned and appropriately scopes the work's limitations. My assessment remains a positive weak accept.

**Key Questions For Authors:**

- The VQGAN codebook uses 8192 entries at d=32, yet no utilization statistics are reported. What fraction of entries are actively used at convergence?
- The absence of prior work does not preclude comparisons against general architectures, direct spectrogram regression, seq2seq without VQ, or a conditional diffusion model. Can the authors provide at least one such baseline?
- Can the authors provide a quantitative spindle or slow-wave detection metric comparing ground-truth and synthesized EEG?
- Does the learned respiratory–EEG mapping generalize to other standard clinical channels (e.g., F3–M2, O1–M2)?
- Can the authors comment on whether synthesized EEG is expected to support more sensitive clinical tasks?

**Limitations:**

Yes

**Strengths And Weaknesses:**

Strengths:

- The asymmetric embedding design is empirically validated, heavier respiratory encoders degraded performance, supporting the minimal linear projection choice
- The masked generative objective is justified by good ablations, compared with AR
- Downstream task evaluation uses an identical ViT backbone across all conditions, cleanly isolating gains attributable to cross-modal translation
- Sleep staging improvement over current SOTA on the same benchmark datasets is practically meaningful
- Demonstrates generative translation across physiological domain boundaries (pulmonary → neurological), with a creative asymmetric embedding design that is non-obvious from prior work
- Demonstrates EEG synthesis from contactless RF reflections, opening a new direction for ambient neurological monitoring

Weaknesses:

- Codebook uses 8192 entries at d=32, which seems low; no codebook ablations or utilization statistics are reported, leaving codebook collapse as an unaddressed risk
- No baselines are provided for the reconstruction task — no diffusion models, no seq2seq transformers without VQ tokenization, no direct spectrogram regression; without these it is difficult to attribute performance to architectural design vs. dataset scale
- Primary reconstruction metric (MAE on normalized spectrograms) may not capture physiologically meaningful structure — transient events like sleep spindles and slow waves have direct clinical relevance but are not explicitly evaluated
- Single-channel (C4–A1) scope is acknowledged but not motivated; EEG-based diagnosis typically relies on spatial patterns across channels, and it is unclear whether the respiratory–EEG coupling generalizes beyond global sleep states to channel-specific signals
- Downstream tasks evaluated are relatively coarse proxies; whether synthesized EEG supports more clinically sensitive tasks (epilepsy, narcolepsy, psychiatric biomarkers) is untested

---

> ### Author Rebuttal · Authors · 2026-03-31
>
> Thank you for the careful and constructive evaluation. We address your concerns below using new experiments and analyses, and hope these revisions justify an increased score.
>
> **Codebook ablations and utilization statistics**
>
> We ablated the tokenizer across multiple axes:
>
> - *Codebook size:* Reducing from 8,192 to 1,024 entries increased L1 error by 5%.
> - *Token size:* The 32×8 patch (~4 Hz × 4 min) isolates canonical EEG bands into distinct tokens. Replacing it with a 256×4 patch (collapsing the full frequency axis) increased L1 error by 12.7%. Finer resolutions (e.g., 16×4) were computationally prohibitive due to the increased self-attention cost.
> - *Discriminator:* Switching from PatchGAN to StyleGAN with R1 regularization reduced L1 by 10% and stabilized training.
> - *Learning rate:* Cosine annealing vs. constant rate showed no significant difference.
> - *Codebook utilization:* At tokenizer convergence, utilization was 100%, facilitated by L2 normalization of latent codes during quantization (Appendix A.2.1). Generator-side utilization was 79.8%, confirming the model draws on a diverse vocabulary rather than collapsing to a subset of entries.
>
> We will add these results to the revised paper.
>
> **No reconstruction baselines**
>
> We implemented a regression baseline: a seq2seq Transformer generating EEG spectrograms pixel-by-pixel in continuous space without tokenization. This increased average MAE by 40% (from 0.067 to 0.094 on the normalized spectrogram) and produced visibly blurrier spectrograms that wash out transient features. Combined with our architectural ablations — sequential autoregressive (+21.5% MAE), column-wise autoregressive (+14.8%), and fully masked training (+4.7% MAE) — these results confirm that performance stems from deliberate design, not dataset scale alone.
>
> **Capturing transient features like spindles and slow waves**
>
> Figure 4 qualitatively shows the model preserves the transient features the reviewer highlights. To quantify this, in the absence of explicit labels, we computed Pearson correlations between synthesized and ground-truth EEG in the slow-wave band (0–4 Hz) and spindle band (12–16 Hz), yielding 66% and 55% respectively. These correlations measure how faithfully the temporal dynamics of band-specific power are preserved across the night. We note that spindles are brief, stochastic events (~0.5–2 seconds) being reconstructed at a 4-minute token resolution, making exact temporal alignment inherently impossible; yet the model still captures the majority of spindle-band power dynamics.
>
> Additionally, the inter-subject variance analysis shows the model preserves 63–77% of within-stage individual standard deviation (vs. ≈0% for a stage-average template), confirming it captures fine-grained structure beyond gross spectral patterns. The narcolepsy and Alzheimer's results below provide further functional evidence: these conditions produce specific EEG abnormalities that would be invisible to a model capturing only coarse patterns.
>
> **More clinically sensitive downstream tasks**
>
> We added two neurological conditions classically assessed via sleep EEG: narcolepsy (AUC: Syn-EEG 0.76, GT-EEG 0.76, breathing 0.67) and Alzheimer's detection (AUC: Syn-EEG 0.75, GT-EEG 0.76, breathing 0.72). Synthesized EEG performs near ground-truth levels and outperforms breathing alone.
>
> We also clarify the clinical relevance of our original tasks: (1) Age estimation is an established EEG-specific task where the brain age index predicts dementia (Gallagher et al., 2025), mortality (Brink-Kjær et al., 2022), and neuropsychiatric disease (Paixao et al., 2020). (2) Sleep staging is a core EEG clinical task. (3) Sex classification validates that the model preserves hormone-mediated spectral signatures, given documented sex differences in spectral power and spindle characteristics (Carrier et al., 2001; Fernandez & Lüthi, 2020).
>
> **Single-channel scope and generalization to other channels**
>
> We chose C4–A1 because it is the standard clinical derivation for sleep EEG, optimally capturing the features foundational to our downstream tasks — slow-wave activity, sleep spindles, and vertex waves. Crucially, the respiratory signal is a single, spatially unresolved measure of global pulmonary effort, which inherently limits recovery of channel-specific spatial information such as anterior-posterior gradients. C4–A1 is well-suited to this input, reflecting global sleep neurophysiology. The framework can extend to multi-channel settings (e.g., a 3D tokenizer over time × frequency × channel), but the respiratory signal's spatial limitation remains a fundamental constraint. We hypothesize that central derivations will perform comparably to C4–A1, while channels sensitive to localized patterns (e.g., occipital alpha) may show larger reconstruction gaps.
>
> We hope these analyses address the reviewer's concerns and strengthen the contribution.

---

> > ### Author Rebuttal · Reviewer_81qV · 2026-04-03
> >
> > All five of my concerns have been adequately addressed. In light of this, I will maintain my positive score.

---

> > > ### Author Response · Authors · 2026-04-06
> > >
> > > We sincerely thank the reviewer for confirming that all five concerns have been fully resolved and are grateful for the constructive and specific feedback throughout the review process. We will integrate the new results and clarifications into the paper.

---

### Official Review · Reviewer_Uwyt · 2026-03-13

**Soundness:** 2
**Presentation:** 3
**Significance:** 4
**Originality:** 3
**Overall Recommendation:** 4
**Confidence:** 5

**Summary:**

EEG contains clinically useful information but collecting it is laborious for both participants and researchers. On the other hand, respiration signals are much easier to acquire through wearable belts or contactless RF sensing. This paper studies a novel problem of generating sleep EEG from respiration. Specifically, the authors propose translation of raw respiration time series to discrete EEG spectrogram tokens. The model is trained on a large, multi-dataset corpus spanning 14 datasets, 28,394 individuals, and 33,919 nights. The experiments evaluate (1) reconstruction quality and (2) if the synthesized EEG is useful for downstream tasks such as age estimation, sex prediction, and sleep staging.

**Compliance With Llm Reviewing Policy:**

Affirmed.

**Final Justification:**

"The authors have successfully addressed my technical concerns. While the experimental results are both interesting and compelling, the manuscript still lacks a robust scientific discussion to ground these findings. Specifically, the relationship between the observed variables remains largely empirical, and more effort is needed to explain these links within a broader scientific context. On the strength of the empirical evidence, I am recommending a Weak Accept.

**Key Questions For Authors:**

- Can the authors provide ablations to empirically justify the main representation choices? Particularly, understanding the role of spectrogram-based EEG targets and discrete VQ tokens would be most helpful, since they are central to the proposed method.
- Can the authors provide more detailed scientific hypothesis behind the relationship between EEG and respiration? Furthermore, I'd like to see "scientific discussions" of the author's interpretation of the relationship based on the observed results.
- Can the authors elaborate on “growing evidence of respiratory–neurological coupling” and explicitly state the physiological hypothesis behind the task? Particularly, it would help to discuss what aspects of EEG the authors believe are recoverable from respiration signals.
For sleep staging, can the authors report macro-F1 and/or confusion matrices in addition to overall accuracy?
- Can the authors test the synthesized EEG on more EEG-relevant downstream tasks? Sleep staging is the most convincing task in the current paper. Additional EEG-specific tasks like that would strengthen the claim that the synthesized signal preserves uniquely **EEG-relevant** information. For instance, some tasks could be abnormality detection or seizure detection; these are just examples, but the point is that testing on tasks relevant to EEG would give more clarity on the quality of the synthesized EEG data.
- Can the authors clarify which downstream datasets were also part of translation training data? This would help readers understand transfer to entirely unseen data better.
- Can the authors more directly justify why they chose the sampled masking schedule? Did you compare against training with fully masked EEG targets throughout? If so, was the benefit mainly in the translation task, the downstream evaluations, or both?'

Minor comments on writing:
- Are the “physiology as language”, “biological grammar” framings intended as a metaphor, or as a central technical claim? If central, the paper would benefit from supporting evidence. Otherwise, I would suggest softening the “language” and “grammar” framing.
Change “28,000 individuals” to “28,394 individuals” in abstract.

**Limitations:**

Yes

**Strengths And Weaknesses:**

Strengths:

- The problem being studied is practically important. If useful EEG information can be synthesized from respiration, this could help EEG-specific analyses in settings where EEG collection is impractical.
- The cross-physiology translation from pulmonary to neurological signals is interesting and appears meaningfully different from the within-domain physiological translation examples discussed in prior work.
- The authors report translation metrics across multiple cohorts with different demographics (age, gender, race) and clinical conditions. This helps readers be more careful about applying the model more broadly.
- The scale of the study is a strength. Specifically, training on 28,394 individuals should help generalize across individuals in many different tasks.
- Among the evaluation tasks, sleep staging is the most convincing. It provides evidence that the synthesized EEG preserves information useful for an EEG-relevant task.

Weaknesses:

Overall, I think the paper addresses an interesting and meaningful problem, but the current experiments do not yet justify several central design choices strongly enough, and the downstream evaluation does not fully establish EEG-specific fidelity.

- Several key design choices are under-justified either empirically or theoretically. Most importantly, the paper does not empirically justify the use of spectrogram-space EEG targets instead of waveform-space targets, or the use of discrete VQ tokens instead of continuous targets. These choices are central to the method, but they are not isolated experimentally. This is especially noticeable because of wording such as “with the asymmetric representations established,” which suggests a level of evidence that the paper does not yet provide. To a lesser extent, the same applies to the use of raw respiration signals, linear projection instead of CNN on respiration, and the sampled masking schedule.
- While the task is well-motivated, the scientific hypothesis underpinning the work remains underdeveloped. The authors point to respiratory–neurological coupling, but the manuscript lacks an explicit hypothesis explaining why respiration would contain sufficient information to synthesize meaningful EEG data. Furthermore, the authors do not sufficiently discuss their results to explain the relationship in a scientific context. Without this theoretical grounding and discussions, the paper reads more like a technical implementation than a rigorous scientific endeavor.
- The downstream task selection is mixed in how it validates EEG fidelity. Sleep staging is well aligned with the paper’s goal as it is directly related to the EEG domain. In contrast, age and sex prediction are weaker tests of whether the synthesized signal captures uniquely **EEG-specific** information. These tasks are not meaningless, but they can plausibly be predicted from multiple physiological signals, including respiration itself.
- The downstream comparison does not fully isolate the gain. Synthesized EEG and ground-truth EEG are evaluated in spectrogram form, whereas the breathing baselines are evaluated in raw form; this makes it difficult to determine how much of the gap reflects recovery of EEG-relevant information versus advantages of the target representation. This is especially important while interpreting the age/sex tasks.
- Since sleep stages are heavily imbalanced, overall accuracy alone is hard to interpret. Reporting macro-F1 would be much more informative of the actual performance.
- Several downstream evaluations appear to draw from the same datasets used during translation training. This could limit the testing of transfer to entirely unseen datasets. A separate table on downstream datasets not used in translation training would strengthen the generalization claim.

---

> ### Author Rebuttal · Authors · 2026-03-31
>
> We thank the reviewer for the thorough evaluation and for rating significance as "excellent." We conducted substantial new work to address the concerns. We hope it justifies raising the score.
>
> **Why EEG spectrograms?**
> This is because:
>
> - *Clinical Sleep EEG is Frequency-Defined:* Unlike short clinical EEGs focused on waveforms, sleep EEG is defined in the frequency domain via spectral analysis:
>
>  - *Sleep Staging:* AASM rules use frequency bands (delta for slow-wave, sigma for spindles), making spectrograms ideal (Prerau et al, 2017).
>
>   - *Psychiatric Biomarkers:* Spectral shifts are key diagnostic/predictive markers for depression (Steiger et al, 2015), PTSD (Denis et al, 2021), and schizophrenia (Ferrarelli et al, 2010).
>
>   - *Pharmacology & Cognition:* Spectral features decode cognitive health (Sun et al, 2023) and track CNS drug effects on sleep architecture (Steiger et al, 2015).
>
> - *Raw EEG is Unsuitable for Generative Models:* Time-series EEGs are stochastic; clinical content relies on power spectral density, not instantaneous phase (Prerau et al, 2017). Predicting raw waveforms would burden the model with reconstructing exact voltage trajectories whose instantaneous phase is stochastic and mainly uninformative.
>
> **Why discrete tokens?**
>
> - Empirically, a continuous seq2seq Transformer regression baseline predicting pixel-by-pixel spectrograms increased average MAE by 40% (6.7% to 9.4%) and yielded blurrier outputs, proving tokenization's value.
>
> - Motivated by computer vision (Yu et al, 2024), discrete tokens enhance Transformer performance. Converting regression to codebook classification prevents generating implausible EEG patterns and reduces mode averaging, preserving sharpness.
>
> **Scientific hypothesis for our work**
>
> Strong evidence supports bidirectional respiration-brain coupling during sleep via 3 mechanisms:
>
> - Respiratory entrainment of cortical oscillations. Nasal breathing entrains oscillations in the piriform cortex, amygdala, and hippocampus (Zelano et al, 2016). Breathing is a timing signal modulating cortical excitability (Heck et al, 2017). Crucially, respiration aligns with sleep-specific oscillations (slow oscillations, spindles, coupled complexes)—the core building blocks of sleep EEG (Schreiner et al, 2023).
>
> - Shared brainstem regulation. The pre-Bötzinger complex generates respiratory rhythms and projects to the locus coeruleus and mediodorsal thalamus, regulating central arousal (Yackle et al, 2017). Sleep depth/stage are accompanied by well-documented changes in respiratory rate and waveform
>
> - Sleep reshapes respiratory patterns. Brain states continuously sculpt breathing (Yackle et al, 2017): deep NREM induces slow, regular breathing; REM causes irregularity; arousals trigger transient disruptions.
>
> While this coupling is established, literature doesn't prove full sleep EEG spectrograms can be reconstructed from respiration alone. However, AI routinely discovers latent physiological relationships: diagnosing Parkinson’s from nocturnal breathing (Yang et al, 2022), predicting 130 conditions from a single sleep PSG (Thapa et al, 2026), identifying kidney disease/diabetes from retinal images (Zhang et al., 2021), and detecting hyperthyroidism (Lin et al, 2024) alongside hundreds of phenotypes from ECGs (Ulloa-Cerna et al, 2025) despite lacking prior predictive links.
>
> **Downstream tasks**
>
> We added two neurological tasks: narcolepsy (AUC: Syn-EEG 0.76, GT-EEG 0.76, breathing 0.67) and Alzheimer’s detection (AUC: Syn-EEG 0.75, GT-EEG 0.76, breathing 0.72).  We also clarify the clinical relevance of our original tasks: 1) Age estimation is an established EEG task where the gap between predicted and actual age (the brain age index) predicts dementia (Gallagher et al, 2025), mortality (Brink-Kjær et al, 2022), and neuropsychiatric diseases (Paixao et al, 2020). 2) Sex classification validates hormone-mediated spectral signatures (Carrier et al, 2001); and 3) sleep staging is a core EEG task. Across all five tasks, synthesized EEG substantially outperforms breathing, confirming the translation captures clinically meaningful neural information.
>
> **Leakage:**
> There is no data leakage. Consistent patient-wise 4-fold cross-validation is strictly maintained across the VQGAN, translation model, and downstream tasks. This guarantees synthesized EEGs evaluated downstream are exclusively generated from breathing inputs unseen during training.
>
> **F1 for sleep staging:** 0.76 (Syn-EEG), 0.81 (GT-EEG)
>
> **Justifying masking schedule:**
> Following Li et al. 2023, variable masking ratios unify generation and representation learning. High masking encourages realistic generation; lower masking captures co-occurrent structures among EEG tokens and between EEG/breathing. Empirically, 100% masking raised MAE by 4.7%
>
> **Representation confound.**
> We replaced breathing waveform with spectrogram; it worsened performance (Age error: 14; Sex AUC: 0.70)
>
> **Writing.** Thanks. We will update as suggested.

---

> > ### Author Rebuttal · Reviewer_Uwyt · 2026-04-02
> >
> > I thank the authors for their detailed responses; my technical questions have been answered. I expect these clarifications to be fully integrated into the camera-ready version.
> >
> > To further improve the paper, I strongly suggest that the authors explicitly clarify that the translation from respiration to EEG is valid only during sleep. Currently, the title and introduction are framed in a way that could be perceived as slightly misleading. For example, I'd recommend:
> >
> > Title: 'Translating Respiration to EEG during Sleep.'
> >
> > Research Question: 'Can we synthesize high-fidelity EEG from breathing alone during sleep?'
> >
> > I encourage the authors to emphasize this scope throughout the manuscript where appropriate. Pending these adjustments, I am happy to increase my recommendation to a Weak Accept."

---

> > > ### Author Response · Authors · 2026-04-06
> > >
> > > We sincerely thank the reviewer for the careful re-evaluation and for confirming that the technical concerns have been fully resolved. We greatly appreciate the constructive engagement throughout this process, which has materially strengthened the paper.
> > >
> > > We agree with the suggestion regarding scope clarity; the translation from respiration to EEG is indeed specific to sleep, and this should be unambiguous. We will adopt the reviewer's recommended framing in the camera-ready version:
> > >
> > > - Updating the title to explicitly reference sleep
> > >
> > > - Adjusting the research question and introduction accordingly
> > >
> > > - Reviewing the manuscript to ensure this scope is clear wherever relevant
> > >
> > > We will also integrate the new results, ablations, and clarifications from the rebuttal into the camera-ready. We are grateful for the reviewer's thoroughness and for the suggestions that improved both the scientific rigor and the clarity of the manuscript.

---

### Decision · Program_Chairs · 2026-04-30

**Decision:**

Accept (regular)

**Comment:**

The paper proposes a new cross-physiology task - that of generating sleep EEG spectrograms from nocturnal breathing signals. The approach is validated on the primary task of reconstruction and downstream tasks of age estimation, sex detection, and sleep staging. The approach is expected to positively impact contactless neurological monitoring. The paper has received split reviews with reviewers in favor of accepting the paper (4,4,5) and one for rejecting it (2).


The reviewers found the problem being studied to be practically important and the cross-physiology generation task itself a contribution. The technical approach is seen to be creative and non-obvious from previous work. Various aspects - the asymmetric embedding design, cross-modal learning, masked generative objective - are appreciated. The scale of experimentation is seen to be a strength. The experiments are conducted with care and the improvement over the SOTA, especially for sleep staging, convincing and practically meaningful as a proof of concept for non-wearable technology for sleep monitoring.

While most reviewer concerns seem to have been addressed in the author-reviewer discussions, the dissenting reviewer (fWH6) has a strong concern that the problem is not sufficiently well-defined and that respiratory signals may not be sufficiently informative of EEG-derived sleep dynamics in a scientifically meaningful fashion. And, as a logical extension, direct estimation without EEG reconstruction may suffice for downstream tasks. The concern is that only spurious statistical correlations might be getting learned – this has precedence in healthcare.

I understand the concerns, and subject to an additional layer of whetting, cautiously recommend acceptance. ML learning of spurious patterns actually points to lack of generalization and proper testing protocols which can test for the same. This is for the ML community to figure out.